# Cryo-EM structure of native human thyroglobulin

Ricardo Adaixo [1,5], Eva M. Steiner[2,5], Ricardo D. Righetto[1,5], Alexander Schmidt [3], Henning Stahlberg [1,4 ✉] & Nicholas M. I. Taylor [2 ✉]

The thyroglobulin (TG) protein is essential to thyroid hormone synthesis, plays a vital role in the regulation of metabolism, development and growth and serves as intraglandular iodine storage. Its architecture is conserved among vertebrates. Synthesis of triiodothyronine ($T_3$) and thyroxine ($T_4$) hormones depends on the conformation, iodination and post-translational modification of TG. Although structural information is available on recombinant and deglycosylated endogenous human thyroglobulin (hTG) from patients with goiters, the structure of native, fully glycosylated hTG remained unknown. Here, we present the cryo-electron microscopy structure of native and fully glycosylated hTG from healthy thyroid glands to 3.2 Å resolution. The structure provides detailed information on hormonogenic and glycosylation sites. We employ liquid chromatography–mass spectrometry (LC-MS) to validate these findings as well as other post-translational modifications and proteolytic cleavage sites. Our results offer insights into thyroid hormonogenesis of native hTG and provide a fundamental understanding of clinically relevant mutations.

[1] Center for Cellular Imaging and NanoAnalytics, Biozentrum, University of Basel, Mattenstrasse 26, 4058 Basel, Switzerland. [2] Novo Nordisk Foundation Center for Protein Research, University of Copenhagen, Blegdamsvej 3B, 2200 Copenhagen, Denmark. [3] Proteomics Core Facility, Biozentrum, University of Basel, Klingelbergstrasse 72, 4058 Basel, Switzerland. [4] Present address: Lab. of Biol. Electron Microscopy, Institute of Physics, SB, EPFL, and Department of Fund. Microbiol., Faculty of Biology and Medicine, UNIL, 1015 Lausanne, Switzerland. [5] These authors contributed equally: Ricardo Adaixo, Eva M. Steiner, Ricardo D. Righetto. ✉email: henning.stahlberg@epfl.ch; nicholas.taylor@cpr.ku.dk

Thyroglobulin (TG) is a 660 kDa hyperglycosylated protein expressed in thyrocytes and secreted to the follicular lumen where it accumulates[1]. Dimeric TG is secreted to the follicular cavity and iodinated to different extents at specific tyrosine residues, a process modulated by the dietary iodine intake[2]. Iodinated TG is transported to the thyrocyte cytosol by pinocytosis and digested, releasing triiodothyronine (T3) and thyroxine (T4) hormones[3]. TG is simultaneously a precursor for thyroid hormone (TH) biogenesis, the synthesis of which is itself modulated by thyroid-stimulating hormone (TSH) signaling[1], and the carrier protein responsible for iodine storage in the follicle colloid. THs are essential to fetal and infant brain development as well as metabolism regulators throughout adulthood[4]. Mutations in the TG sequence, or alteration of glycosylation structures, are related to increased risk of thyroid cancer as well as dyshormonogenesis associated with goiter[5–7].

Analysis of primary sequences of TG allowed early identification of internal homology domains classified as type 1, type 2, and type 3 repeats[8], as well as a cholinesterase-like domain (ChEL) at the carboxyl end of the protein[9]. Type 1 repeats occur 11 times in the human TG (hTG) sequence and are homologous to 1a domains found in other proteins with known structure[10,11]. The ChEL domain also has several homologs with structures determined by X-ray diffraction[12–15]. The ChEL domain assists TG folding, dimerization, and secretion processes[16].

Despite the extensive biochemical characterization of TG in the past decades[1,17–22], the three-dimensional structure of TG remained unknown[1,3], until the full-length structure of human thyroglobulin was revealed recently, providing a much-improved understanding of its structure and function[23]. Here, we present the structure of endogenous, fully glycosylated hTG determined by cryogenic transmission electron microscopy (cryo-EM). The obtained EM map depicts a dimer with extensive interchain contacts which include but are significantly larger than the ChEL dimer interface. We observe 57 disulfide bridges (DSB) per hTG monomer, which add structural stability, solubility, and rigidity to most of the protein[1,3,23]. However, the extreme N- and C-terminal segments as well as two other regions (areas which we here term as "foot" and "wing") display a higher degree of flexibility, which is likely related to function.

We provide a comprehensive structural description of the endogenous hTG dimer and demonstrate the functional importance of the natural post-translational modification and iodination sites by presenting an atomic model of the nearly complete protein. The native hTG sample is heterogeneous both in composition and conformational states, which likely represents the in vivo requirements for hTG function.

While this work was in preparation, Coscia et al.[23] described the structure of endogenous, deglycosylated hTG from thyroid glands of patients with goiters (3.4 Å global resolution) as well as recombinant, non-deglycosylated hTG (3.7 Å global resolution), identified the putative hormonogenic sites and validated them using an in vitro hormone production assay. Our findings are consistent with those of Coscia et al. but our cryo-EM reconstruction (3.2 Å global resolution) is of the native, non-deglycosylated hTG from patients without known thyroid deficiencies and of relatively better structural resolution.

## Results

We obtained a homogenous solution of hTG from native sources, originating from healthy human thyroid glands, by resuspension and gel filtration of the commercially available lyophilate without performing any in vitro iodination. The cryo-EM images of plunge-vitrified hTG solution displayed monodisperse particles in different orientations with size and shape consistent with

previous observations in negative stain preparations[17]. A cryo-EM map at 3.2 Å nominal resolution was reconstructed allowing atomic modeling of the hTG dimer to around 90% completeness (2,483 modeled residues over 2,748 expected residues per chain) (Supplementary Tables 1 and 2). Poorly resolved parts of the map, in particular at the N-terminal and other flexible parts, prevented modeling of the full-length protein. This composite map is the result of a globally refined consensus map and two maps locally refined around particularly flexible regions, the "wing" (3.8 Å resolution) and the "foot" (4.1 Å resolution), as described below (Fig. 1 and Supplementary Movie 1). These peripheral and flexible domains are likely an obstacle to obtaining diffraction-quality crystals, which probably prevented the structural determination of hTG in the past. The hTG dimer is ~250 Å long by 160 Å wide and 110 Å along the C2 symmetry axis. Each chain is formed by regions I, II, and III and a C-terminal ChEL domain (Fig. 1). The interface between monomers buries an area of 31,100 Å² involving all regions except region II. The hTG structure is annotated similarly to what is reported in the literature[1], however, region I lacks the so-called "linker" between repeats 1.4 and 1.5. We identified 57 DSBs over the overall hTG structure (Supplementary Fig. 1 and Supplementary Table 3) stabilizing the protein and three types of cysteine-rich internal homology repeats in hTG: type 1, type 2 and type 3 (Supplementary Table 2). There are 10 type 1 repeats within region I and an 11th in region II, three type 2 repeats, each bearing 14 to 17 residues, lie between the hinge region and repeat 1.11. Type 3 repeats are located between repeat 1.11 and the ChEL domain.

**Type 1 repeats**. The first four type 1 repeats in the proximal region I cluster at the N terminus of each chain and establish extensive contacts with region III and the hinge of the opposing chain, as well as intrachain contacts with repeat 1.5 (Figs. 1 and 2). Within repeats 1.1 to 1.4, we observe two non-homologous insertions (NHI), namely on loops 2 and 3 of repeat 1.3. Repeats 1.5 to 1.10 occupy the central core of hTG and this ensemble forms contacts to all other regions on both chains. Additional NHIs are present in loop 1 of repeat 1.5, loop 2 of repeat 1.7, and loop 2 of repeat 1.8 (Supplementary Fig. 7). Insertions of repeats 1.3 and 1.5 are in proximity and exposed between the proximal region I and the ChEL domain of the opposing chain. Insertions of repeat 1.8 from both chains lie at the C2 symmetry axis and form a helix bundle providing an additional 1,710 Å² interchain contact surface (Fig. 3). Repeat 1.7 exposes an NHI protruding almost radially to the C2 symmetry axis and forms no additional contacts. We named this protrusion as "foot", and it is flexible as suggested by the diffuse density obtained in the consensus map. This region was resolved to 4.1 Å after local refinement (see "Methods" section for details). Residues 378 to 615 were assigned to the so-called "linker region" in previous work while here they form the insertion of repeat 1.5. A consequence of our annotation is that repeat 1.5 encompasses 2 additional cysteines, Cys408 and Cys608, and a total of 4 disulfide bridges, therefore we classify this repeat as type 1c, as opposed to type 1a and type 1b repeats containing 6 and 4 cysteine residues respectively. The remaining NHIs are devoid of additional cysteine residues.

**Type 2 repeats**. hTG residues 1456 to 1487 comprise three contiguous type 2 repeats flanked by the hinge and repeat 1.11. Each type 2 repeat comprises 2 cysteine residues, all of which are engaged in disulfide bond formation (Fig. 4). The most N-terminal cysteine of each type 2 repeat establishes a disulfide bond (DSB) with the adjacent N-terminal domain while the

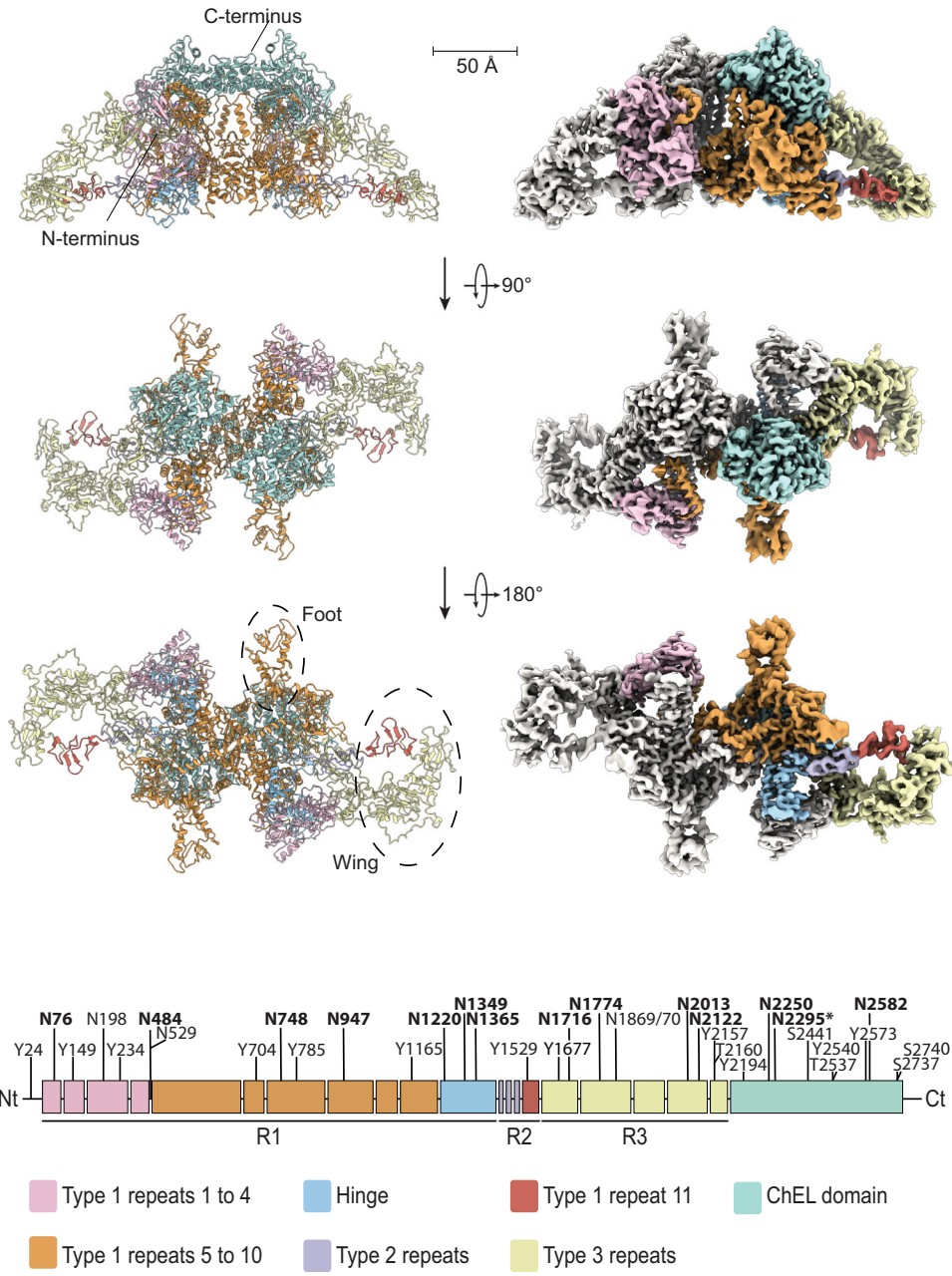

**Fig. 1 The cryo-EM map of hTG at 3.2 Å.** (Right) Density map of hTG with one monomer in white and corresponding atomic model (left). Map and model are colored as in the bottom linear diagram of the primary structure of hTG.

most C-terminal cysteine establishes a DSB with the adjacent C-terminal domain. Therefore, both repeat 1.11 and hinge region are linked to repeats 2.1 and 2.3 via DSBs contrary to previous hypothesis[8] where all 6 cysteines in type 2 repeats would form DSBs internally.

Repeat 2.1 has a small beta strand between both cysteine residues which is the only secondary structure element found within the set. All type 2 repeats have a shape reminiscent of an arrowhead where the pointy edge of one repeat is embraced by the flat base of the following type 2 repeat.

It has been hypothesized that the CXXC motif in type 2 repeats constitutes a thioredoxin box which may be required for TG multimerization via intermolecular DSBs[24]. We did not investigate this hypothesis. However, all type 2 repeats are extensively solvent-exposed and therefore could serve as a potential substrate to thioredoxin.

**Type 3 repeats**. Region III comprises a total of 5 type 3 repeats linking the ChEL domain to repeat 1.11. These repeats are formed by an alpha helix followed by a three-stranded beta sheet and can be subdivided into type 3a, bearing 8 cysteines, and type 3b, bearing 6 cysteines (Fig. 5). The loop connecting the third beta strand to the neighbor domain is longer and apparently disordered. We noticed the previous annotation[1] places the limits of each type 3 repeat within secondary structure elements and therefore does not take into consideration the globular nature of individual domains which can be discerned in the structure, but, for consistency, we follow the same annotation.

Repeats 3a3 and 3b2 were reasonably well-defined in the consensus map (see "Methods" section) however this was not the case for repeats 3a2, 3b1, 3a1 and down to repeat 1.11. Both 2D classes and consensus refinement maps were not well-defined in the 3a2 to 1.11 region (which we named "wing region") likely due

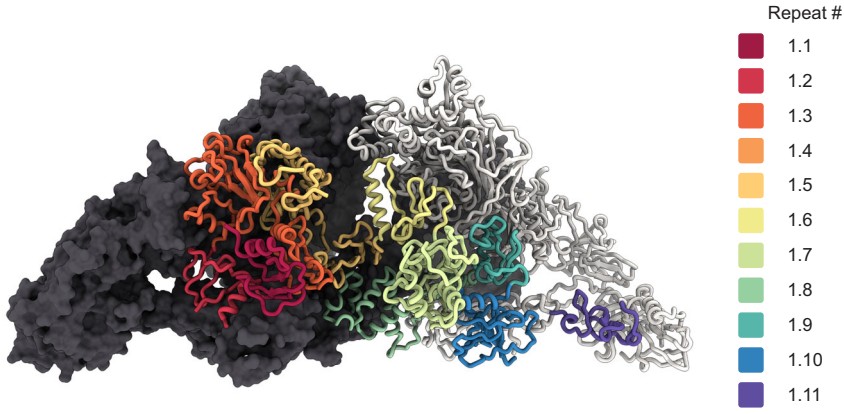

**Fig. 2 Disposition of the different type 1 repeats in the hTG structure.** Surface representation of one hTG monomer in the background-colored charcoal. Ribbon representation of the second hTG monomer in the foreground with each type 1 repeat colored as in the scheme on the right: non-type 1 repeats are colored in light gray.

to increased flexibility. The wing has a "C" shape where the tips seem to act as pivot points, connecting repeats 2.3 to 1.11 at the N terminus and repeats 3a2 to 3b2 at the C terminus.

**Mapping of hTG's hormonogenic sites.** The post-translational iodination of hTG contributes both to thyroid hormonogenesis as well as iodine storage. One hTG monomer contains 66 tyrosines, therefore the hTG dimer represents a huge reservoir for iodination and post-translational modifications. Under sufficient iodide intake 10–15 tyrosine residues become mono- and diiodotyrosines (MIT, DIT), serving as functional hormonogenic units within the hTG structure[3]. Hormonogenesis requires a selected pair of donor and acceptor tyrosine residues and four of such sites (A–D sites) were proposed in hTG[21,23] (Table 1). Coupling occurs between a DIT and another DIT (donor and acceptor) or between a MIT and a DIT (MIT donor and DIT acceptor) to undergo an oxidative quinol-ether coupling reaction to form T4 or T3, respectively[20].

To map the iodination status and positions of hTG, we used liquid chromatography–mass spectrometry (LC–MS) and detected 11 iodinated tyrosine residues that could serve as potential donor/acceptor site for TH production (Fig. 1, Table 1, "Methods" section, and Source Data). Out of the detected sites, 10 were MIT, 6 sites MIT and DIT, 3 only MIT, and 1 site DIT only. A special role, as highlighted before, plays Y24, the most efficient T4 forming unit, where we detected a thyroxine T4 modification as well[21] (Table 1).

We did not observe convincing densities for iodine in the cryo-EM map, probably because iodination levels were low, these sites may be particularly sensitive to radiation damage during data collection, the iodination pattern may be inconsistent between different particles, or a combination of all these factors.

Despite a large number of iodination sites, hormonogenesis depends primarily on tyrosine residues near the N terminus (the A-site with acceptor at Y24, crucial for T4 synthesis) and the C terminus (the C-site with acceptor at Y2766), the main site of T3 formation. Further proposed hormonogenic sites are site B, with acceptor Y2573[18] (iodination detected by MS) and site D (acceptor Y131)[23] (no iodination detected by MS) (Fig. 6 and Table 1). In case of the A-site, although we do not directly observe the acceptor Y24 in the map we can deduce its approximate location as the first modeled residue is P30, which we then use as a proxy for estimating the distance to the putative donor sites. Furthermore, we revealed Y24 to be clearly mono- and

diiodinated and as well showing an additional T4 mass (Table 1). Potential donor sites for Y24 have been suggested to be Y234 (donor 1, distance to P30: 19.9 Å) or Y149 (donor 2, distance to P30: 20.6 Å)[23]. We detected mono and di-iodination for both donor sites Y149 and Y234 providing further evidence supporting that these residues are potential donors within hormonogenic site A[23]. Taken together with the distance to P30 and flexibility of the N-terminal, this suggests that T4 synthesis can occur within a single hTG monomer.

Y258 (which is also close to a glycosylation site N484) has been suggested as an alternative donor. In the structure, this residue is still located relatively close to P30 (distance: 35.3 Å) and therefore also to Y24, however, major conformational changes would be necessary given that the side chain of Y258 is oriented toward the hTG core. At the B-site, Y2573 is located at the surface of the protein and fairly accessible. The residue is proposed to be an acceptor whereas Y2540 functions as a donor (Fig. 6). For both donor and acceptor tyrosines, we observed mono and di-iodination by mass spectrometry, and the 6 Å close contact would allow a coupling reaction. In 15 Å proximity is as well Y2478 but it has not been shown to be iodinated in our sample. Interestingly, T2537, at 3.8 Å and 5.6 Å distance to both tyrosines, Y2573 and Y2540, at the B-site, was found to be phosphorylated and therefore may be playing a crucial role in the acceleration of hormone production[25]. At about 28 Å distance from this T3 production site, S2441 was detected by mass spectrometry to be sulfonated (Table 1). Sulfonated serines can be involved in various functions including protein assembly and signaling processes and this type of PTM was detected in a cathepsin-C-like enzyme from parasites[26].

The C-site is not visible in the structure, where T2727 is the last residue traced. It has been shown that Y2766 is an acceptor to the Y2766 in the neighboring monomer of the hTG dimer[27]. Our model is perfectly consistent with this observation: the C-α distance between the T2708 residues of both monomers is only 20 Å.

The D-site, with a captor Y1310, is accessible and exposed. The donor residue has been proposed to be Y108[23]. We did not observe iodination of Y1310 nor Y108 in the MS data nor in the cryo-EM map. However, since those residues are surface exposed, in vitro iodination was shown to be possible[23].

A wide variety of other iodinated sites (Y785 MIT/DIT, Y1165 MIT, Y1529 MIT, Y2194 MIT/DIT) have been identified and

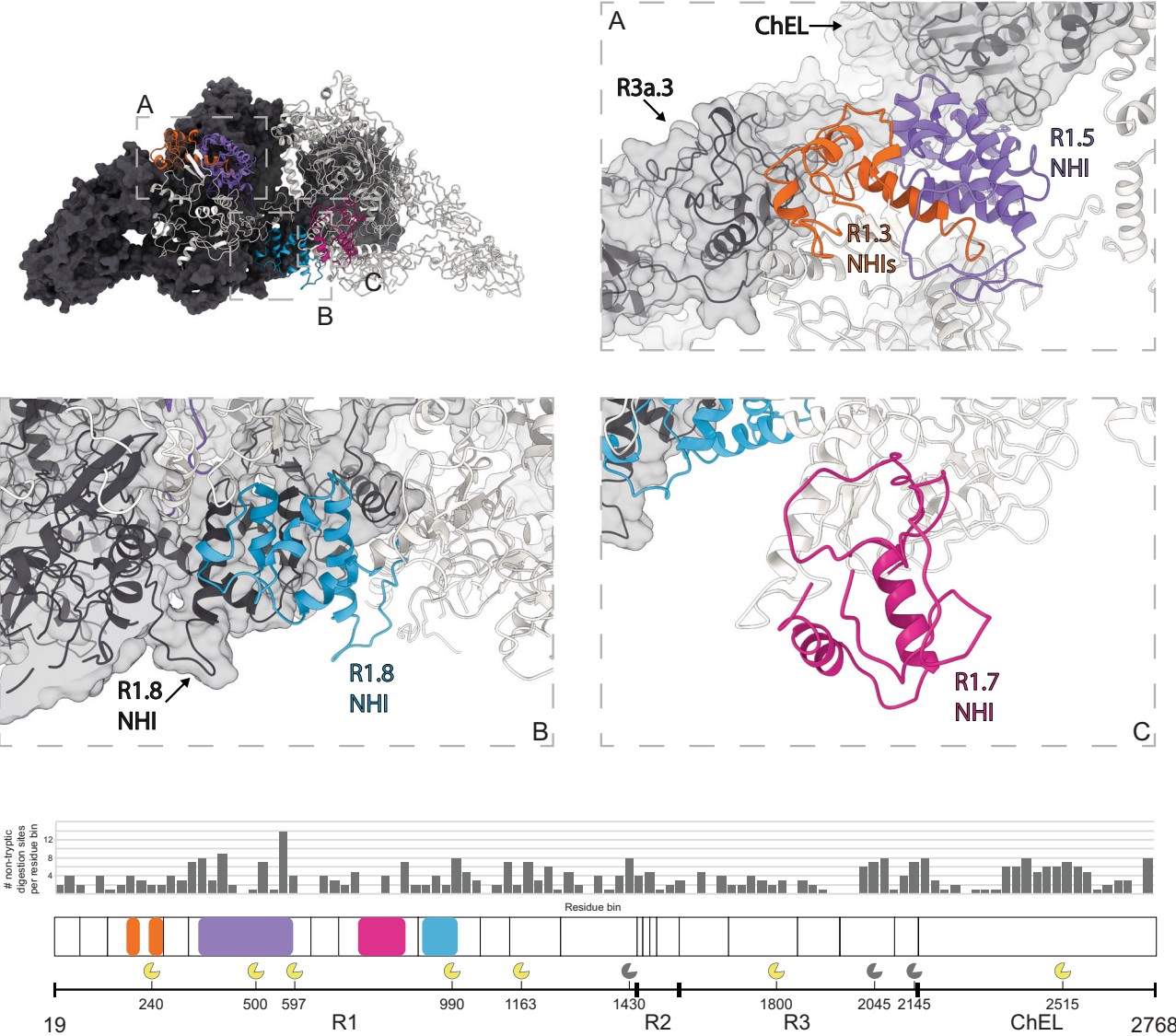

**Fig. 3 Location of type 1 repeat NHIs and proteolysis clusters.** (Top) Surface representation of one hTG monomer (in the background) colored charcoal. Ribbon representation of the second hTG monomer in the foreground with each NHI colored differently: orange—repeat 1.3 NHIs; purple—repeat 1.5 NHI; blue—repeat 1.8 NHI; wine—repeat 1.7 NHI. Detailed clipped views of each insertion are displayed in the dashed boxes; box **B** view direction is the same as the top left image while boxes **A** and **C** were reoriented for better depiction. (Bottom) Histogram of the non-tryptic cleavage sites detected by MS with major sectors depicting the approximate position of previously reported (yellow, see text for references) and herein experimentally determined (gray) cleavage clusters.

most of these are located on the surface (Table 1). However, it is unclear to what extent they have a role in hormonogenesis as opposed to iodine storage.

**Mapping of proteolysis sites**. The lifecycle of TG comprises multiple proteolysis events: the cleavage of the N-terminal 19-residue signal peptide; the N- and C-terminal cleavages which liberate iodopeptides; the limited proteolysis of TG, which releases TG particles from the colloid agglomerate; and, finally, the digestion of TG internalized in the thyrocyte[29–31]. Cathepsins, a family of cysteine proteases, perform the mentioned proteolytic attacks on TG both inside the thyrocyte and in the follicular lumen. The approximate locations of cathepsin proteolysis sites are depicted in Fig. 3. Digestion of the extreme N- and C-terminus containing thyroid hormones (TH) is among the earliest proteolysis events experienced by mature TG[28–30].

Insertions of repeats 1.3, 1.5, and 1.8 contain motifs targeted by cathepsins and a sequence of proteolysis events has been suggested[31] involving different proteases from the cathepsin family. Hence, one function of these insertions is to expose proteolysis-prone segments in order to facilitate hTG digestion.

To further understand the proteolysis patterns in hTG, we analyzed our sample for proteolysis using LC–MS (Fig. 3). Our hTG atomic model lacks the residues between N496-P547 and T1781-N1814 because no clear density was observed in those regions in the EM map. Interestingly, two of the proteolysis sites plotted in Fig. 3 (major sector marks on positions 500 and 1800) lie within these missing segments, suggesting that our sample was partially digested at these specific locations.

**hTG glycosylation and other modifications**. The addition of glycan structures to hTG is crucial for protein folding, structure,

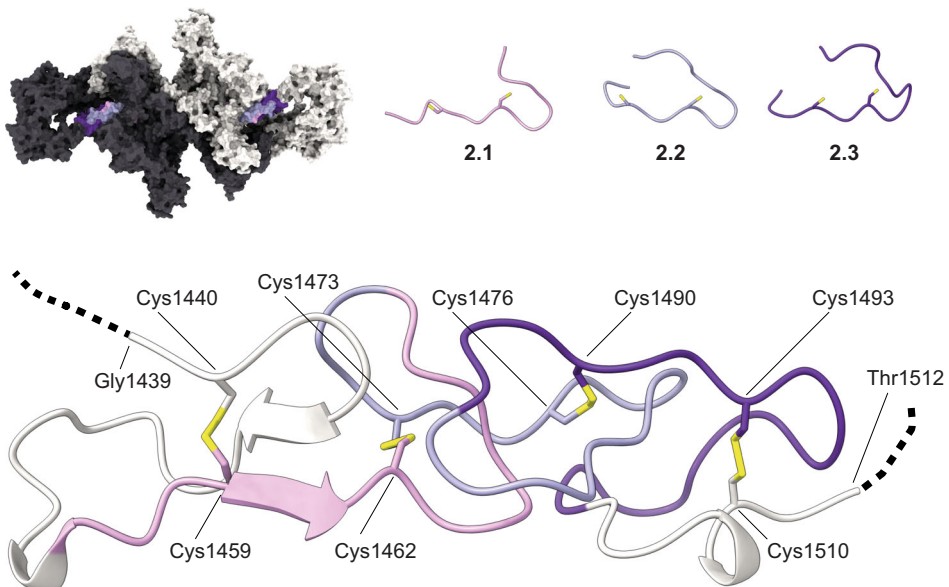

**Fig. 4 Type 2 repeats.** (Top) Representation and location of the type 2 repeats as pink, light, and dark purple ribbon within the hTG structure and independent repeat structures. (Bottom) Ribbon representation of the type 2 repeats as disposed in the context of the hTG structure. Cysteine residues are represented in sticks with DSBs and sulfur atoms colored yellow.

and therefore function, as well as immune-recognition and cell signaling, and plays a significant role in protein transport and THS production[6]. Within the human thyroglobulin monomer 16 N-linked glycosylation sites have been discovered in the mature protein[19,23].

We identified 16 N-linked glycosylation sites, per hTG monomer, by mass spectrometry (Fig. 1 and Table 1). Out of those, 14 asparagine residues showed an additional density in the map and N-linked glycan structures were modeled (Supplementary Fig. 8). The two glycosylation sites, N198 and N529, are located close to the hormonogenic A-site and within the region of a proteolytic site, respectively, and therefore not modeled (Fig. 1). Based on the map an additional glycosylation site at position N2295 was identified and a high mannose was modeled indicating that there might be more than 16 sites in the mature hTG protein (Table 1)[23].

We present previously unknown glycosylation sites at position N198, N1869, and N2122, that have not been described in previous biochemical studies, although N1869 and N2122 have been observed structurally (Table 1)[19,23]. For several of the previously annotated glycosylation sites we did not detect any modification nor did we observe any additional map density; N110, N198, N816, and N1348[19,23]. Besides glycosylation, we have also mapped sulfonation, phosphorylation, and acetylation. We identified 15 acetylation sites, 4 phosphorylation, and 4 sulfonation sites (Table 1). No methylations or succinations were detected.

The phosphorylation at T2537 ($PO_4$) and sulfonation Y2540 ($SO_3$) are within the hormonogenic B-site. Phosphorylation is thought to improve the efficiency of T3 formation[25] and sulfonation of Y24 and the surrounding peptide sequence was shown to be crucial in thyroid hormone synthesis[32,33].

Sulfated iodotyrosines (Tyr-S) have a short lifespan before the coupling reaction occurs and it is suggested that after Tyr-S binding to peroxidase, where it is iodinated, the sulfate group is removed, releasing an iodophenoxy anion available for coupling with an iodotyrosine donor[33]. For Y2540, the donor residue in the B-site, we detected sulfonation, MIT and DIT representing all 3 states of hormone site preparation and a perfectly prepared hormone formation site (Fig. 6).

**Mapping of nonsense and missense mutations.** We considered all the nonsense and missense mutations reported previously[1,34], and plotted these on the hTG structure (Fig. 7). The majority of mutations causing early termination of hTG translation as well as those causing a change in amino acid identity fall within modelled regions; 37 of 42 nonsense mutations and 120 of 130 missense mutations. Interestingly, 4 of the 5 major clusters of the mentioned mutations (Fig. 7) overlap with the proteolysis sites depicted in Fig. 3, namely those within repeats 1.3, 1.8, 1.10, and to a lesser extent the ChEL domain.

**Comparison with structures of deglycosylated endogenous hTG and natively iodinated bovine Tg.** We provide a comparison between the model obtained in this study with the recent model published by Coscia et al. (Supplementary Fig. 9). Both models are overall in close agreement, however certain portions of the structure, in particular at the "wing" and the "foot" are now better resolved and in some regions deviate from the previous assignment by Coscia et al. We additionally describe post-translational modifications of hTG (Table 1).

While this work was in revision, Kim et al. reported the structure of natively iodinated bovine thyroglobulin[35]. Interestingly, they observed three modified tyrosines with additional densities in their structure, indicative of acceptor sites: Y24 ($T_4$), Y2041 (DIT), and Y2575 ($T_4$) (Supplementary Table 4). We did observe thyroxination of the unmodeled Y24 residue by MS (as well as MIT and DIT). Y2041 is not conserved in hTG. We detected modification of hTG Y2573 (corresponding to bovine TG Y2575) to MIT and DIT by MS in hTG, but no additional density could be observed in our cryo-EM map for this residue. Kim et al. also reported four tyrosines where side-chain density was lacking despite clear main chain density: Y108, Y149, Y1395, Y2542 (Supplementary Table 4). This is indicative of donor sites. We did however observe side-chain density for both hTG Y149 as well as Y2540 (corresponding to bovine TG Y2542). As for the other residues, hTG Y108 could not be modeled and bovine TG Y1395 is not conserved in hTG. We also compare opposing cysteine pairs and formed disulfide bridges between our hTG structure and that of bovine TG. Kim et al. observed several pairs

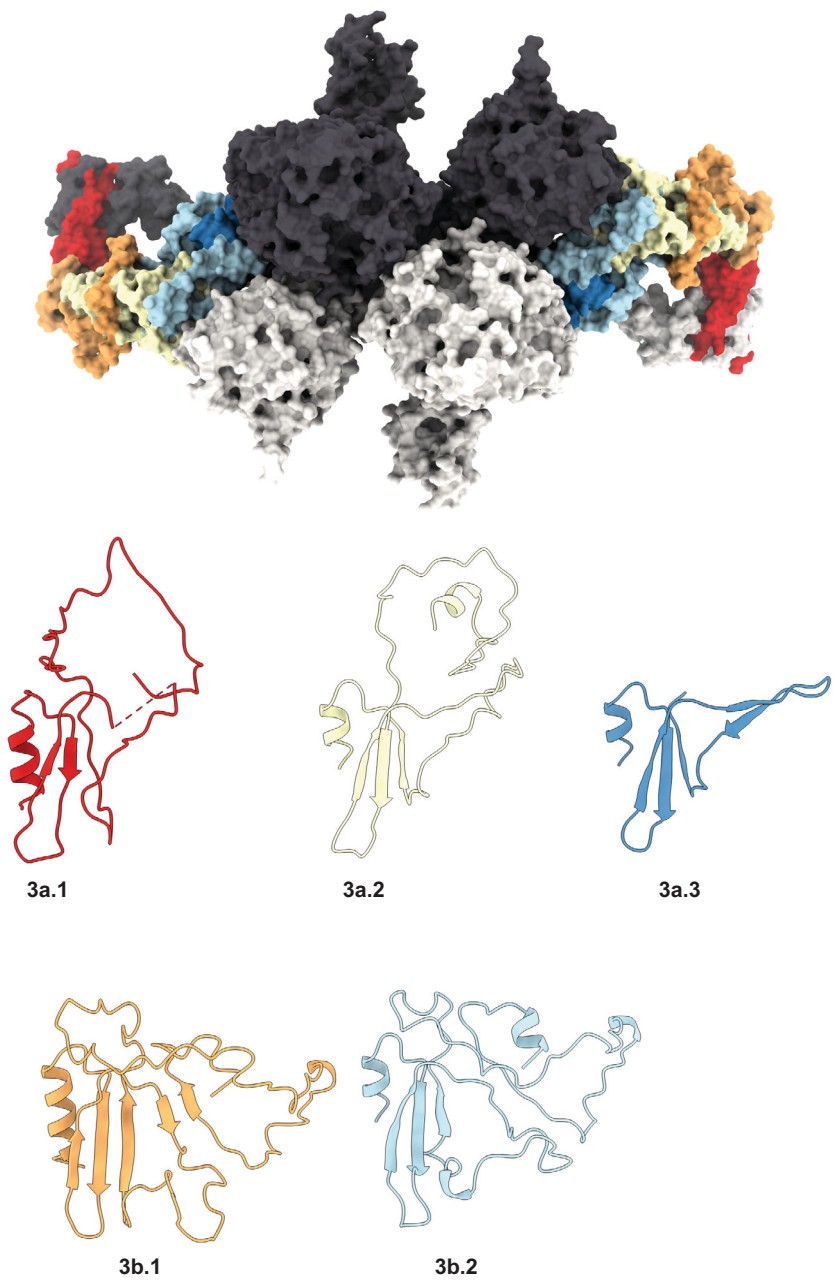

**Fig. 5 Type 3 repeats.** (Top) Surface representation of the hTG dimer with type 3 repeats colored as represented in the bottom panel; the other individual hTG structures within a monomer are highlighted charcoal and light gray. (Bottom) Ribbon representation of the aligned type 3 repeats.

of cysteine residues that were in close proximity (~3 Å), but which did not form disulfide bridges (Supplementary Table 3). We do not see any evidence for such non-formed disulfide bridges in our hTG map, but the overall resolution of our structure is lower than that for the bovine TG structure.

### Discussion
We determined the atomic structure of hTG based on a composite cryo-EM density map at an overall resolution of 3.2 Å. The overall dimeric structure is consistent with previous biochemical experiments, which showed that the cholinesterase domain is necessary and sufficient for hTG dimerization, at least when overexpressed in HEK293 cells[36], but also reveals the participation of regions 1 and 3 in dimer contact formation.

The ChEL domains and repeats 1.6–1.8 form the core of the dimer and lie near the C2 symmetry axis while regions 2 and 3, as

well as the remaining type 1 repeats, occupy more peripheral zones. We modeled 2 of the expected 4 hormonogenic sites, namely site B and site D. Acceptor tyrosines of sites A and C are located at the extreme N- and C-termini of the hTG chain and could not be modeled likely due to the inherent flexibility of these regions.

Our structural characterization also evidences features of hTG that were previously unknown: the environment and possible function of the 4 NHIs present in type 1 repeats; the previously annotated linker region which in fact is a NHI of repeat 1.5; the globular nature of type 3 repeats, which could be annotated differently, and finally the flexible nature of the wing and foot regions.

All NHIs are solvent-exposed and present peptide motifs recognized by proteases as determined by LC–MS analysis. The NHI of repeat 1.7 is unique in the sense that the remaining NHIs

**Table 1 List of human thyroglobulin modifications identified by LC–MS.**

| Modification site[a] | Type of modifications identified | Modification previously known[a] |
|---|---|---|
| Y24 | MIT, DIT, T$_4$ | Yes all |
| N76 | N-Glycosylation | Yes |
| Y149 | MIT, DIT | Yes all |
| N198 | N-Glycosylation | Yes |
| Y234 | MIT, DIT | No all |
| N484 | N-Glycosylation | Yes |
| N529 | N-Glycosylation | Yes |
| Y704 | MIT | Yes |
| N748 | N-Glycosylation | Yes |
| Y785 | MIT, DIT | Yes, no |
| N947 | N-Glycosylation | Yes |
| Y1165 | MIT | No |
| N1220 | N-Glycosylation | Yes |
| N1348 | N-Glycosylation | Yes |
| N1349 | N-Glycosylation | Yes |
| N1365 | N-Glycosylation | Yes |
| Y1529 | MIT | No |
| Y1677 | DIT | No |
| N1716 | N-Glycosylation | Yes |
| N1774 | N-Glycosylation | Yes |
| N1869 | N-Glycosylation | No |
| N2013 | N-Glycosylation | Yes |
| N2122 | N-Glycosylation | No |
| Y2157 | O-Sulfonation | No |
| T2160 | Phosphorylation | No |
| Y2194 | MIT, DIT | No |
| N2250 | N-Glycosylation | Yes |
| N2295 | N-Glycosylation | Yes |
| S2441 | O-Sulfonation | No |
| T2537 | Phosphorylation | No |
| Y2540 | MIT, DIT, O-Sulfonation | No all |
| Y2573 | MIT, DIT | Yes all |
| N2582 | N-Glycosylation | Yes |
| S2737 | Phosphorylation, O-Sulfonation | No all |
| S2740 | Phosphorylation | Yes |

*MIT* monoiodotyrosination, *DIT* diiodotyrosination, *T$_4$* thyroxine.
[a]The amino acid numbering system presented is including signal peptide residues 1–19, following Uniprot: P01266 numbering (Date of download: 20 April 2020).

establish contacts to adjacent domains other than the type 1 repeat itself. One speculative hypothesis is that repeat 1.7 NHI could still form interdomain contacts but in the context of hTG multimerization.

The hTG atomic structure is decorated with a variety of post-translational modifications which were further studied by LC–MS. Importantly, we describe three previously unreported glycosylation sites. The density of detected proteolysis sites and the multiple iodination states found for Y24 strongly indicate that our hTG sample was heterogeneous.

We expect that the hereby-presented structure of native hTG leads to an improved understanding of TG biology which can be applied in the diagnosis and therapy of thyroid disease, where our model could be valuable to determine the location of different antibody epitopes and their relation to autoimmune diseases.

## Methods

**Sample preparation.** Human thyroglobulin (catalog no. T6830; Sigma-Aldrich) was dissolved in gel filtration buffer (25 mM Tris-HCl pH 7.5, 150 mM NaCl, 1× sodium azide) and injected onto a Superdex-200 increase size-exclusion chromatography column connected to an ÄKTA purifier FPLC apparatus (GE Healthcare Bio-Sciences). Peak fractions were pooled and concentrated to 2 mg/mL of protein before plunge freezing.

**Cryo-EM sample preparation and data collection.** Quantifoil 2/2 400 mesh Cu grids were glow discharged in low-pressure air for 30 s. Then 3 µL of concentrated hTG were dispensed to the hydrophilic surface of the grid prior to single side blotting for 2 s and plunge freezing in liquid ethane using a Leica EM GP2 plunger (Leica Microsystems) operating at 20 °C and 80% relative humidity.

Frozen grids were imaged using a Titan Krios (ThermoFisher Scientific) transmission electron microscope operating at 300 kV equipped with a Gatan Quantum-LS energy filter (slit width 20 eV; Gatan Inc.) and a K2 Summit direct electron detector (Gatan Inc.) (Supplementary Table 1). SerialEM[37] was used for automated data collection with 7 acquisitions per hole using beam-image shift[38]. Movies were recorded in counting mode with a pixel size of 0.64 Å/px at the sample level. Each movie comprised an exposure of 50 e−/Å$^2$ fractionated into 50 frames over 10 s.

**Image processing and model building.** Movies were preprocessed online in FOCUS[39] using MotionCor2[40] for drift correction and dose weighting and

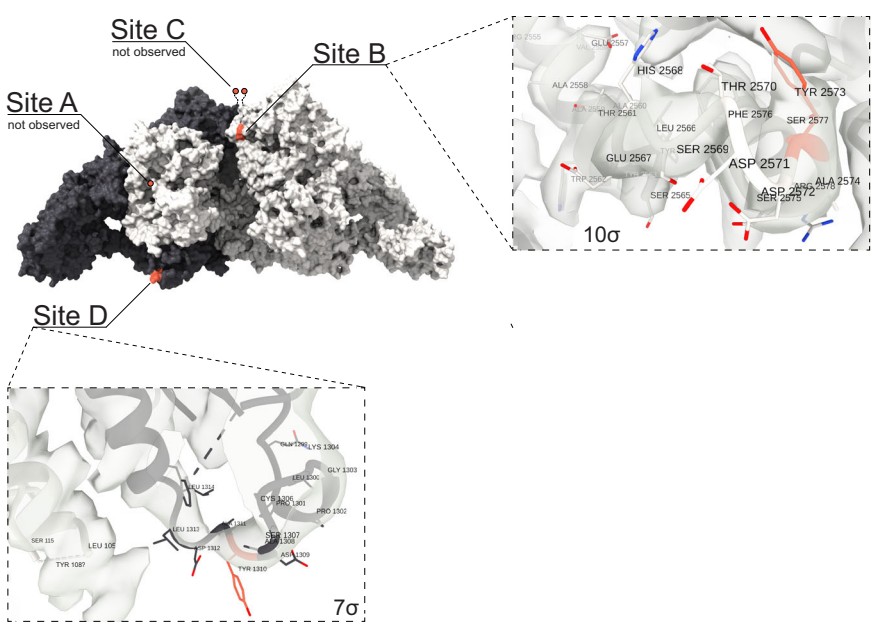

**Fig. 6 Location of hormonogenic sites in hTG structure.** (Top left) hTG surface representation with one monomer in charcoal and one monomer in white. Acceptor tyrosine residues, Y2573 and Y1310 in ~15 Å proximity[18] at each canonical site (A to D) are marked in red, for which at Y2573, additionally MIT and DIT were detected in MS experiments. The insets show sites B and D in detail with the respective cryo-EM density and its corresponding threshold level to aid confidence assessment.

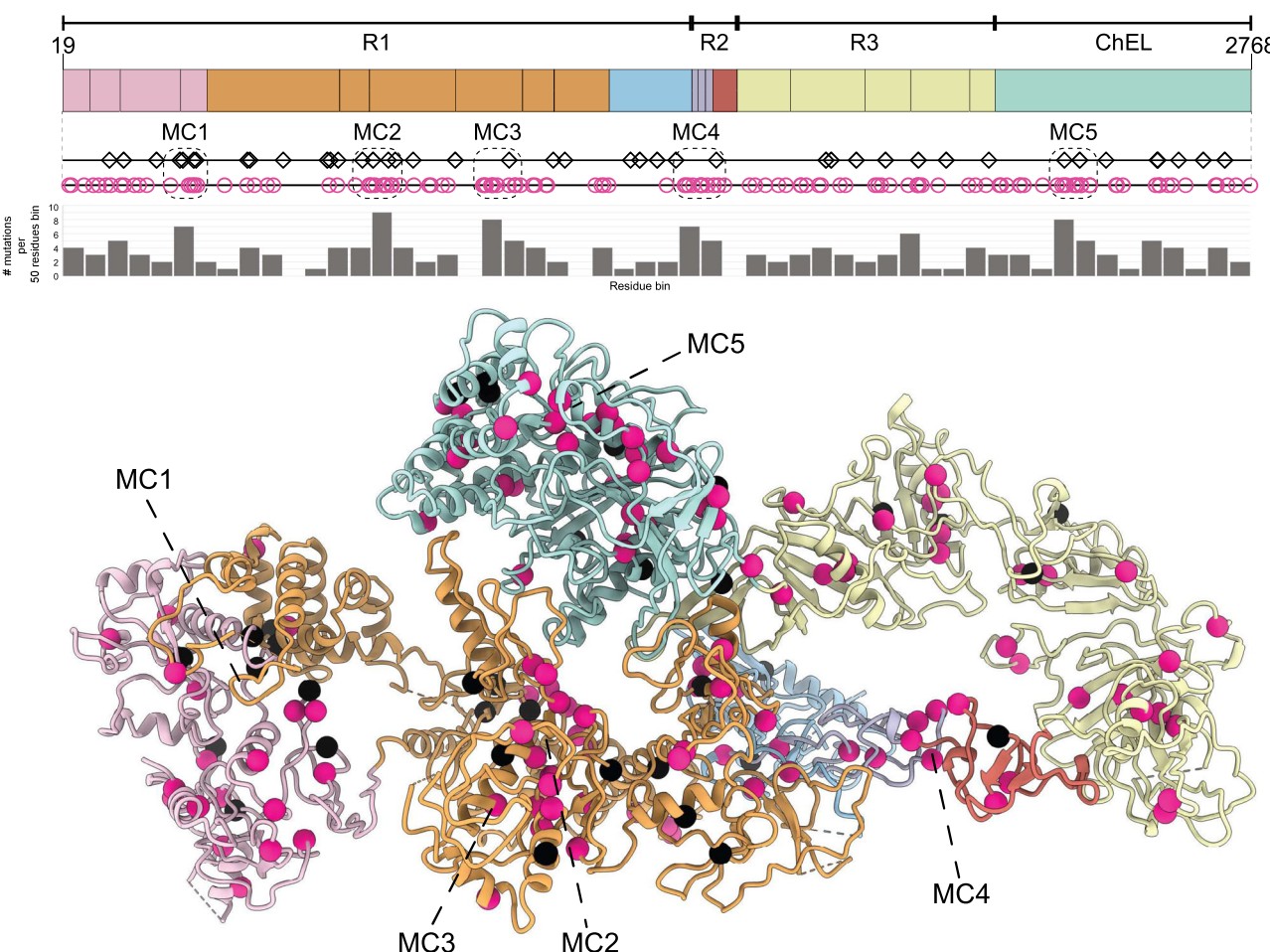

**Fig. 7 Location of nonsense and missense mutations in hTG responsible for congenital hypothyroidism.** (Top) hTG linear diagram depicting the location of nonsense mutations in black squares and missense mutations in pink circles[34]; domain color code is the same as Fig. 1. Dashed boxes represent the 5 mutation clusters (MC1 to MC5) with the highest density of mutations. Bar diagram visualizing the number of mutations per 50 residues. (Bottom) Location of the same mutations in the ribbon representation of hTG; nonsense mutations in black spheres and missense mutations in pink spheres.

CTFFIND 4.1[41] for contrast transfer function estimation. Out of 8,119 movies acquired, 4,504 had an estimated CTF resolution better than 4 Å and were selected and used for automated particle picking in Gautomatch (Zhang, K., http://www.mrc-lmb.cam.ac.uk/kzhang/) with a CC threshold of 0.4 using a Gaussian blob as a template.

Particles were classified in 2D using RELION-3[42] and the best classes were selected for ab initio map generation and auto-refinement. EMAN2[43] was subsequently used to create projections of the refined map. Template-based automated particle picking in Gautomatch was then applied using 20 Å low-pass filtered projections as templates. This picking was run on 7,266 movies with an estimated CTF resolution better than 6 Å. The new set of particles was pruned by 2D and 3D classification resulting in 37,619 particles being allocated to one class with well-defined features and apparent C2 symmetry. This class was further refined imposing C2 symmetry, and particles were corrected for beam-induced motion and CTF refined in RELION-3 (Supplementary Fig. 2). A consensus map with a nominal resolution of 3.3 Å based on the FSC curve at 0.143 criterion[44,45] was obtained after post-processing using an automatically estimated B-factor of −45 Å² (Supplementary Figs. 3 and 4).

Particles considered in the consensus map were imported into cryoSPARC v2[46] for localized refinement with the aim of improving the quality of the densities in the "wing" and "foot" regions. Masks around these regions were created in UCSF Chimera[47] using a local resolution filtered version of the consensus map as a template and the volume segmentation tool. Both regions benefited from the local refinement procedure as the resulting maps display better connectivity and side-chain densities, compared to the consensus map in the considered regions (Supplementary Figs. 2, 5, and 6). Interestingly, the best local refinement maps of the wing region were obtained without performing any prior signal subtraction, as judged by visually inspecting the densities.

An atomic model of hTG based on the consensus, "foot" and "wing" maps were manually built de novo in Coot[48]. Model building of the cholinesterase domain was started by making a homology model using HHPred[49] based on the structure of human acetylcholinesterase inhibited by the (−) stereoisomer of VX (PDB ID

6CQT)[50]. The final hTG model covers 90% of the amino acid sequence, lacking mainly loops and the N- and C- terminal extensions. Focused maps and half-maps were merged using the program phenix.combine_focused_maps and post-processed in RELION like the consensus map. The atomic model was real-space refined in PHENIX[51] and validated using phenix.mtriage[52] and MolProbity[53].

**LC–MS analysis**. 30 μg of human thyroglobulin were dissolved in 0.8 M urea (Sigma), 0.1 M ammonium bicarbonate by vortexing and ultra-sonication (5 s, Vial Tweeter, Hielscher, Germany). Then, the protein was reduced with 5 mM TCEP for 60 min at 37 °C and alkylated with 10 mM chloroacetamide for 30 min at 37 °C. After splitting the sample into 3 aliquots containing 10 μg of protein each, samples were digested by incubation with sequencing-grade modified trypsin, sequencing-grade Glu-C, and sequencing-grade chymotrypsin (all 1/50, w/w; Promega, Madison, Wisconsin), respectively, overnight at 37 °C. After acidification using 5% TFA, peptides were desalted on C18 reversed-phase spin columns according to the manufacturer's instructions (Macrospin, Harvard Apparatus) and dried under vacuum.

The setup of the μRPLC-MS system was as described previously[54]. Chromatographic separation of peptides was carried out using an EASY nano-LC 1000 system (ThermoFisher Scientific), equipped with a heated RP-HPLC column (75 μm × 37 cm) packed in-house with 1.9 μm C18 resin (Reprosil-AQ Pur, Dr. Maisch). Phosphorylated peptides were analyzed per LC–MS/MS run using a linear gradient ranging from 95% solvent A (0.1% formic acid) and 5% solvent B (80% acetonitrile, 19.9% water, 0.1% formic acid) to 30% solvent B over 60 min at a flow rate of 200 nL/min. Mass spectrometry analysis was performed on Q-Exactive HF equipped with a nanoelectrospray ion source (both ThermoFisher Scientific). Each MS1 scan was followed by high-collision-dissociation (HCD) of the 20 most abundant precursor ions with dynamic exclusion for 20 s. Total cycle time was approximately 1 s. For MS1, 3e6 ions were accumulated in the Orbitrap cell over a maximum time of 100 ms and scanned at a resolution of 120,000 FWHM (at 200 m/z). MS2 scans were acquired at a target setting of 1e5 ions, accumulation

time of 100 ms, and a resolution of 15,000 FWHM (at 200 $m/z$). Singly charged ions and ions with unassigned charge state were excluded from triggering MS2 events. The normalized collision energy was set to 28%, the mass isolation window was set to 1.4 $m/z$ and one microscan was acquired for each spectrum.

The acquired raw-files were converted to the mascot generic file (mgf) format using the msconvert tool (part of ProteoWizard, version 3.0.4624 (2013-6-3)). Using the MASCOT algorithm (Matrix Science, Version 2.4.1), the mgf files were searched against a decoy database containing normal and reverse sequences of the predicted UniProt entries of *Homo sapiens* (Uniprot, download date: 7 March 2019) and commonly observed contaminants (in total 41,592 sequences) generated using the SequenceReverser tool from the MaxQuant software (Version 1.0.13.13). The precursor ion tolerance was set to 10 ppm and fragment ion tolerance was set to 0.02 Da. The search criteria were set as follows: semi-tryptic, semi-Glu-C and chymotryptic specificity was required for the corresponding 3 different prepared samples, 3 missed cleavages were allowed, carbamidomethylation (C), was set as fixed modification. The following variable modifications were applied separately for all samples: (i) +16 on M (Oxidation), +42 on Peptide N-Terminal K (Acetyl), +126 on Y (Iodo); (ii) +16 on M (Oxidation), +252 on Y (Diiodo), +378 on Y (Triiodo); (iii) +14 on KR (Methyl), +16 on M (Oxidation), +100 on K (Succinyl); (iv) +16 on M (Oxidation), +80 on STY (Phospho); (v) +1 on NQ (Deamidated), +16 on M (Oxidation), +80 on ST (Sulfo); (vi) +16 on M (Oxidation), +470 on Y (Triiodothyronine), +596 on Y (Thyroxine). Next, the database search results were imported to the Scaffold Q+ software (version 4.9.0, Proteome Software Inc., Portland, OR) and the protein and peptide false identification rate was set to 1% based on the number of decoy hits for each generated dataset. Protein probabilities were assigned by the Protein Prophet program[55]. Proteins that contained similar peptides and could not be differentiated based on MS/MS analysis alone were grouped to satisfy the principles of parsimony. Proteins sharing significant peptide evidence were grouped into clusters. Only peptides of human thyroglobulin that satisfied these strict identification criteria were reported (see Table 1 and Fig. 3).

**Reporting summary**. Further information on research design is available in the Nature Research Reporting Summary linked to this article.

## Data availability
The EM map for the complete hTG molecule has been deposited in the EMDB under accession code EMD-12073. Atomic coordinates for hTG have been deposited in the Protein Data Bank under the accession code PDB 7B75. The mass spectrometry proteomics data have been deposited to the ProteomeXchange Consortium via the PRIDE partner repository with the dataset identifier PXD029013. LC–MS data used to generate tables and figures has been provided as a .xlsx Source Data file (SourceData_PTMs_Ionscore40_final.xlsx). Source data are provided with this paper.

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

## Acknowledgements
This work was funded by the Swiss National Science Foundation (NCCR TransCure) and the Novo Nordisk Foundation (NNF17OC0031006). We thank K. Goldie and L. Kovacik for their support in electron microscopy. Calculations were performed at sciCORE (http://scicore.unibas.ch) scientific computing center at the University of Basel. The Novo Nordisk Foundation Center for Protein Research is supported financially by the Novo Nordisk Foundation (grant NNF14CC0001). E.M.S. acknowledges the support of the Lundbeckfonden through a postdoctoral fellowship. N.M.I.T. is a member of the Integrative Structural Biology Cluster (ISBUC) at the University of Copenhagen.

## Author contributions
R.A. and R.D.R. performed protein purification and cryo-EM sample preparation, data collection, and processing. R.A., E.M.S., and N.M.I.T. built the atomic model. A.S. performed LC–MS analysis. R.A, E.M.S., and R.D.R. analyzed data and interpreted results. H.S. and N.M.I.T. supervised the work. All authors jointly wrote the manuscript and approved its final version.

## Competing interests
The authors declare no competing interests.
