## [Peer Review File · Nature Communications]

Cryo-EM structure of native human thyroglobulinREVIEWER COMMENTS

Reviewer #1 (Remarks to the Author):

This manuscript addresses the cryo-EM structure of human thyroglobulin (hTG) at a resolution of 3.2 Å, which was previously published by Coscia et al., 2020 at a resolution of 3.5 Å. (Coscia et al., 2020 created cryo-EM datasets for both recombinant TG and endogenous TG covering 93% of the hTG.) The structure here presented covers about 90% of the hTG. However, the ultimate source of commercial native hTG here used is not specified (although it is purchased from SIGMA, it is unclear if it comes e.g. from human patients with any kind of thyroid disease, normal controls, etc.). Not having a homogeneous source TG sample might lead to issues in the interpretation of the results. Second, this manuscript utilizes mass spectrometry to analyze posttranslational modifications in hTG. However, this methodology (mass spectrometry) is not described in the corresponding section (nor elsewhere in the manuscript). Further, the results/ raw data of the mass spectrometry is not presented in the manuscript; thus, it is not possible to evaluate the quality of it. Third, the authors claim in the introduction section that hTg monomer has 57 disulfide bridges and it is unclear how this assumption is made from the partial cryo-EM TG structure here presented. Fourth, to examine the location of clusters of nonsense and missense mutations in TG the authors have not used updated data, and thus, the analysis/result is not current. Additional comments are shown below. Last, it is important to note that this manuscript does not properly discuss (nor even cite at times) previous valuable knowledge in the field which lead to overestimation of some of the results here presented.

Major comments

1) The abstract should be improved: 1) It should mention that mass spectrometry is utilized to analyze posttranslational modifications of TG (and which of those in particular); 2) It should list the main findings by each of the techniques employed; 3) The amount of text related to the previously known information and the explanation of what is done in this manuscript and findings should be balanced.

2) Line 79-80 (“Despite the extensive biochemical characterization of Tg in the past decades the three-dimensional structure of Tg remained unknown, limiting the understanding of its function”) should be edited to reflect the current state of the knowledge. Please make sure that it is clear to the reader that the cryo-EM structure of the hTg was published in 2020.

3) Please include a new table and figure indicating the position of the 57 pairs of disulfide bonds that were found in your TG cryo-EM structure (according to line 83 in the manuscript), and the distance between each of the pairs.

4) The source of the hTG used in this manuscript should be further addressed. What is the source of human TG in the commercial Human Thyroglobulin (catalog no. T6830; Sigma-Aldrich)? Does it come from patients with any kind of thyroid disease, normal controls, etc.? To mention only a few potential issues with a source of non-homogeneous TG sample: 1-TSH levels impact on the posttranslational modifications of TG, which could impact on its 3D structure; 2-mutated TG may lead to modifications on its 3D structure. If this was the case, how would the result of the cryo-EM TG structure and mass spec represent those of wild type native TG?

5) Lines 90-94. These paragraph lead to misinterpretation of previously published work in the field. Please re-write to represent the actual TG samples Coscia et used in their paper from 2020. They utilized not only a full-length recombinant human TG (rTG) but also endogenous TG from thyroid glands of patients with goitres (eTG) to create cyo-EM structures.

6) There are several discordances with previous limits (amino acid numbering) stablish in the primary hTg structure and also the cryo-EM hTg published by Coscia. This last two were compared Mol Cell Endocrinol. 2021 May 15;528:111227. doi: 10.1016/j.mce.2021.111227. Epub 2021 Mar 6. I strongly recommend including a new table with the comparison of your limits for each Tg zone (domain or region) to what is already published.

7) Given the potential source of non-homogeneous hTG utilized in this work, I do not recommend introducing new names to any zone in the Tg structure.

8) This manuscript utilizes mass spectrometry to analyze posttranslational modifications in hTG. However, this methodology is not described in the corresponding section (nor elsewhere in the manuscript). From Table 1, I assume the authors used LC-MS. Please include details on how mass spectrometry was performed and the raw data in each of the corresponding sections (the description of the methodology, the proteases used and where they map in the Tg monomer, the equipment used, the chromatograms).

9) Line 163. Replace "coupling is proposed" for "coupling occurs".

10) Line 167. "...donor/acceptor site for TSH production". The concept is wrong. Please replace for "...donor/acceptor site for thyroid hormone production".

11) Line 176. "...acceptor at Y2766, specific for T4 and T3 synthesis) respectively." Please replace for "...acceptor at Y2766, the main site of T3 formation)."

12) Line 172-173. "The iodination pattern may be inconsistent between different particles." I strongly recommend the authors to perform further experiments to determine iodination sites in TG with an homogeneous source of iodinated hTG to clarify this. The iodination of Tyr and utilization of hormonogenic sites varies according to the different factors, e.g. the availability of iodine, the stimulus of TSH.

13) Line 178. Site E is not represented in Fig 6. Please rectify/move the following "(Fig. 6, Table1)". Was site E determined by cryo-EM? Please clarify and if so, please include in Fig. 6.

14) Please clarify the distances between Tyr24 and all proposed donor sites to Tyr24 that are mentioned in this manuscript: Tyr24-Tyr149; Tyr24-Tyr234; Tyr24-Tyr258

15) Line 202. "We did not observe iodination of Y1310 nor Y108 and our data does not support those residues as a D-site" This is not consistent with what is shown in Fig. 6. Please clarify and amend accordingly in the text / Fig. 6.

16) Line 202-203. Need careful revision of the paper cited (25. Berg G et al. 2020), which I was not able to find online. I recommend incorporating the following bibliography and revise the hypothesis related to in vivo/ in vitro utilization of site D: Biochem Biophys Res Commun. 2014 Jul 18;450(1):488-93. doi: 10.1016/j.bbrc.2014.05.144. Epub 2014 Jun 6; J Biol Chem. 2011 Jan 7;286(1):259-69. doi: 10.1074/jbc.M110.159483. Epub 2010 Oct 26.

17) Table 1. It is not clear what "iodination" is referring to, please indicate MIT or DIT accordingly.

18) Please include a legend to Table 1. Indicate that the numbering of the amino acid position in TG refers to TG including the signal peptide and cite the reference sequence.

19) Line 227-258. Clarify the methodology utilized to determine hTG glycosylation and other modifications as well as the source of hTG. Please add detailed description as mentioned in my previous comment #8.

20) How was the mapping of the TG proteolysis sites created? If this is a predictive informatic analysis this should be indicated in the text and in Figure legend #3.

21) Line 253. “We considered all the nonsense and missense mutations reported previously (1. Citterio et al., 2019)” Is this sentence referring to the (nonsense and missense) mutations in TG responsible for congenital hypothyroidism? If so the data in this manuscript represented in Figure 7 should be updated to the most recent compilation of the hTG mutations responsible for congenital hypothyroidism that was published this year -- please refer to Mol Cell Endocrinol. 2021 May 15;528:111227. doi: 10.1016/j.mce.2021.111227. Epub 2021 Mar 6. Additionally, please modify the sentence “We considered all the nonsense and missense mutations responsible for congenital hypothyroidism reported previously” and add the proper citation.

Minor comments

- 1) Line 81. “hTG” replace to “human TG (hTG)”
- 2) Figure legend 1. Line 453. Indicate that you refer to the primary hTG structure.
- 3) Figure legend 2. Clarify what the white/grey ribbon represents.
- 4) Figure legend 4. Indicate the meaning of each color in the ribbons. What does yellow color mean?
- 5) Figure legend 5. Provide further explanation of the representation. What is the upper panel representing? How is it different from the lower panel?
- 6) Figure legend 6. Specify the amino acid number of the acceptor sites. Clarify the methodology by which acceptor sites B and D were identified.
- 7) Supplementary figure S4: Indicate the meaning of FSC (Fourier shell correlation).
- 8) Video. It would be beneficial to match the (letter) color of the names of each of the Tg zones or sites and the respective Tg zone or site in the Tg 3D structure.

Reviewer #2 (Remarks to the Author):

The paper by Taylor and co-workers describes the structure of the native human thyroglobulin using cryo-EM. The paper should be published provided that the points I indicate below are resolved:

- 1 - Provide a comparison to the structure recently published by Coscia et. al., an RMSD and a figure highlighting the major differences found and what the new structures add to the overall picture.

2 - There is no information on model quality and there should be a table with the refinement and validation statistics for the model either in the main text or supplementary info.

3 - The clash score is quite high and the large number of clashes should be addressed.

4 - In all figures, the authors should state and show the densities at 1 σ contour level. This is specially important in Figure S2.

5 - The corresponding half-maps, masks and the raw maps should be deposited to EMDB together with the PDB entry.

6 - Two PDB reports were submitted but correspond to the same entry, please submit one only.

7 - Given the importance highlighted in the manuscript of the glycosylation sites, it would be informative to include a figure showing the density of a representative site.

8 - In figure S1 the local resolution estimate for wing and foot should be provided.

9 - The dip in the FSC curve should be addressed.

10 - Figure S4 the FSC for the consensus map should also be presented. Ln 93 The resolution from the structure published by Coscia et al. should be included. L101 Where are the residues that could not be shown? Ln 123 The estimated local resolution for the foot should be stated. Ln 324 Include the initial model used or if constructed ab initio describe the method. Ln 346 Title is in all caps

Reviewer #3 (Remarks to the Author):

Adaixo et al. present the cryo-EM structure of native human thyroglobulin (Tg). Tg is an important molecular complex that plays a key role in the synthesis of thyroid hormones. In addition to structure

determination, the authors have performed mass spectrometric analysis of their specimen. From these data, the authors gain insight into the residues that are critical for hormone synthesis, and they identify protein residues that are post-translationally modified by iodination, glycosylation, sulfonation, and phosphorylation. Overall, these results are in good agreement with a study that previously reported the structure of native and recombinant human thyroglobulin (Coscia et al., reference 23), though minor differences exist.

One of the major concerns with the study presented by Adaixo et al. is that it mostly ignores the Coscia et al. paper, likely aiming not to compromise the novelty of the findings, even though the Coscia et al. study was published more than one year ago. This not only raises issues with appropriate acknowledgment of prior work, but also leaves the comparison of the new and previous results to the reader, diminishing the utility of the study under consideration.

In its present form, I cannot recommend the manuscript under consideration for publication. The manuscript should be revised to account for the current state of knowledge in the field.

Other comments:

52: "structure conserved among vertebrates" - maybe better use "architecture" or "domain organisation" instead of "structure" unless other structures are known.

90-94: The introduction needs to be rewritten to properly acknowledge and reference prior work by Coscia et al., a paper published in February 2020, i.e. more than one year ago. Doing so will better inform the reader about the current state of the field and the new insight provided by the work under review. The wording suggesting that Coscia et al. reported only the structure of recombinant Tg is inaccurate and should be changed because Coscia et al. reported the structure of human Tg extracted from thyroid glands as well.

204-208: Residues Y883, Y2637, and Y2640 seem to be buried in the protein core and unlikely to be accessible as donors/acceptors. Overall, this section remains unconvincing, particularly because the lack of *in vivo* iodination was used in the preceding paragraph to discount site D, and when also considering the biochemical data in Coscia et al. that indicate the sites A-D account for all hormone production (at least in their system).

238/239: "... that have not been described in previous biochemical studies" - some of these were identified in the structural study by Coscia et al.; while factually correct, this claim appears not to fully convey the current state of knowledge in the field.

Figure 1: It is unclear whether introduction of another new set of domain names is beneficial to the reader. The authors may want to reconsider.

Figure 2: Ribbon representation could use thicker ribbon to make repeats 1.6-1.9 more easily distinguishable against the background and from each other.

Figure 6: The figure shows molecular models of sites B and D, including side chains. Densities supporting these assignments should be shown either directly in this figure or in a supplementary figure. The assignment appears reliable for site B. However, the panel for site D shows residues that are either not built at all in the molecular model (Y108) or whose location is not very well defined due to poor density in this region (Y1310). It is important to convey the level of accuracy and reliability with which these important sites could be modelled to the readers of the paper, preferably in the figure legend and/or in the main text.

Methods: The paper lacks appropriate validation of the cryo-EM-derived molecular model (e.g. refinement statistics, model-map FSC). These need to be added.

Point-by-point response to reviewers' comments

“Cryo-EM structure of native human thyroglobulin” (Adaixo *et al.*, NCOMMS-21-08694)

We would like to thank all the reviewers for their careful reading of our study and for their detailed comments which have been very helpful to further improve the manuscript. We here present a response to all the points raised by the reviewers.

Reviewer #1

This manuscript addresses the cryo-EM structure of human thyroglobulin (hTG) at a resolution of 3.2 Å, which was previously published by Coscia *et al.*, 2020 at a resolution of 3.5 Å. (Coscia *et al.*, 2020 created cryo-EM datasets for both recombinant TG and endogenous TG covering 93% of the hTG.) The structure here presented covers about 90% of the hTG. However, the ultimate source of commercial native hTG here used is not specified (although it is purchased from SIGMA, it is unclear if it comes e.g. from human patients with any kind of thyroid disease, normal controls, etc.). Not having a homogeneous source TG sample might lead to issues in the interpretation of the results.

We thank the reviewer for their thorough assessment and helpful comments.

Human thyroglobulin protein was purchased from Sigma. Upon request of information, the manufacturer specified the source as ‘**The thyroids are from patients without any known thyroid deficiency**’ and ‘**the purification methods are proprietary, but all buffers used are physiological pH**’. The human thyroglobulin used in our study therefore originates from healthy human thyroids glands, natively glycosylated and iodinated and in functional condition. We agree that there might be variation or a spectrum of glycosylation levels or iodination levels, however, because of its source of patients without any known thyroid deficiency it arguably is a more native sample than the one from Coscia *et al.*

Second, this manuscript utilizes mass spectrometry to analyze posttranslational modifications in hTG. However, this methodology (mass spectrometry) is not described in the corresponding section (nor elsewhere in the manuscript). Further, the results/ raw data of the mass spectrometry is not presented in the manuscript; thus, it is not possible to evaluate the quality of it.

Post-translational modifications of this native sample, as mentioned, were investigated using LC-MS experiments. This has now been stated clearly in the text. The methodology for LC-MS experiments has been added to the ‘Methods’ section, and source data for **Table 1** and **Fig. 3** are provided as a .xlsx file (SourceData_PTMs_Ionscore40_final.xlsx). In the section ‘Accession Codes’ we added the information that mass spectrometry proteomics data can be accessed at the ProteomeXchange Consortium webpage

(<http://proteomecentral.proteomexchange.org>) via the PRIDE partner repository with the dataset identifier PXD029013 and 10.6019/PXD029013.

Third, the authors claim in the introduction section that hTg monomer has 57 disulfide bridges and it is unclear how this assumption is made from the partial cryo-EM TG structure here presented.

Indeed, we agree with the reviewer that the number of disulfide bridges we identify per monomer (57) is a lower bound for the total number per monomer, we have modified the text to indicate this.

Fourth, to examine the location of clusters of nonsense and missense mutations in TG the authors have not used updated data, and thus, the analysis/result is not current. Additional comments are shown below. Last, it is important to note that this manuscript does not properly discuss (nor even cite at times) previous valuable knowledge in the field which lead to overestimation of some of the results here presented.

Concerning the nonsense and missense mutations involved in congenital hyperthyroidism, we have now updated **Fig. 7** to include data of Citterio *et al.*, 2021.

Citations and information missing has been added and addressed in the comments below, improving the acknowledgement of prior work and updating the manuscript to the latest knowledge available.

The abstract should be improved: 1) It should mention that mass spectrometry is utilized to analyze posttranslational modifications of TG (and which of those in particular); 2) It should list the main findings by each of the techniques employed; 3) The amount of text related to the previously known information and the explanation of what is done in this manuscript and findings should be balanced.

We have modified the abstract accordingly and incorporated these valuable suggestions.

Line 79-80 ("Despite the extensive biochemical characterization of Tg in the past decades the three-dimensional structure of Tg remained unknown, limiting the understanding of its function") should be edited to reflect the current state of the knowledge. Please make sure that it is clear to the reader that the cryo-EM structure of the hTg was published in 2020.

The work on the hTG structure of Coscia and colleagues and our own work was carried out around the same time, and was completely independent, however our publication process got delayed due to the COVID-19 pandemic. Nevertheless, we acknowledge the importance of this information in the introduction for the reader and have modified the text correspondingly.

Please include a new table and figure indicating the position of the 57 pairs of disulfide bonds that were found in your TG

cryo-EM structure (according to line 83 in the manuscript), and the distance between each of the pairs.

We have added this information to the supplementary data. **Fig. S1** displays all structurally identified 57 disulfide pairs as yellow spheres spreading across the hTg structure. **Table S2** summarizes those disulfide pairs indicating location, cysteine pairs and structural disulfide distances in Å. All S-S distances agree with those expected for disulfide bridges (~2.05 Å). Furthermore, we now include relevant references to prior knowledge about the abundance of disulfide bridges in the introduction in the line mentioning the disulfide bridges.

The source of the hTG used in this manuscript should be further addressed. What is the source of human TG in the commercial Human Thyroglobulin (catalog no. T6830; Sigma-Aldrich)? Does it come from patients with any kind of thyroid disease, normal controls, etc.? To mention only a few potential issues with a source of non-homogeneous TG sample: 1-TSH levels impact on the posttranslational modifications of TG, which could impact on its 3D structure; 2-mutated TG may lead to modifications on its 3D structure. If this was the case, how would the result of the cryo-EM TG structure and mass spec represent those of wild type native TG?

We inquired with the manufacturer for details on the source of human thyroglobulin protein prior to manuscript submission. The manufacturer, Sigma-Aldrich, of the product T6830-5MG stated that protein was extracted from human thyroids of patients without any known thyroid deficiencies. All buffers and methods are proprietary, but all buffers used are at physiological pH.

Therefore, and as mentioned in the above, our structure and data represent details on human thyroglobulin originating from healthy human thyroids glands, natively glycosylated and iodinated. When comparing this to the previously published structural information of Coscia *et al.* 2020, which itself provides insight into glycosylated recombinant hTg (rTg) and deglycosylated endogenous TG from thyroid glands of patients with goitres (eTG), the overall 3D structure very similar only differing in highly flexible regions (see below in our response to reviewer #2), suggesting that differences in PTMS probably do not greatly affect the overall structure of hTG.

Lines 90-94. These paragraph lead to misinterpretation of previously published work in the field. Please re-write to represent the actual TG samples Coscia et used in their paper from 2020. They utilized not only a full-length recombinant human TG (rTG) but also endogenous TG from thyroid glands of patients with goitres (eTG) to create cyo-EM structures.

We thank the reviewer for the valid point. The paragraph was reformulated to better acknowledge the work by Coscia *et al.*

There are several discordances with previous limits (amino acid numbering) stablish in the primary hTg structure and also the cryo-EM hTg published by Coscia. This last two were compared Mol Cell Endocrinol. 2021 May 15;528:111227. doi: 10.1016/j.mce.2021.111227. Epub 2021 Mar 6. I strongly

recommend including a new table with the comparison of your limits for each Tg zone (domain or region) to what is already published.

We thank reviewer #1 for highlighting the differences of domain border zones. As recommended, we prepared a table similar to the recommended article (Citterio *et al.* 2021) comparing the thyroglobulin zone limits between the collective database summary on Uniprot, the previous published structure by Coscia *et al.* 2020 and Adaixo *et al.* (this manuscript). Domain borders are highlighted with colors according to the manuscripts to facilitate easier comparison. The table is provided as supplementary information (**Table S3**).

Given the potential source of non-homogeneous hTG utilized in this work, I do not recommend introducing new names to any zone in the Tg structure.

We respectfully disagree with the reviewer that we introduce new names to zones in the Tg structure when describing our independent structure of fully glycosylated hTG from healthy thyroids (apart from our terms “wing” and “foot”). The domain border limits are highly similar between the provided structures of Coscia *et al.* 2020 and this manuscript. Furthermore, there are no major structural changes or discrepancies between the recombinant, native deglycosylated from goiters (both Coscia *et al.*), and native hTG from healthy human thyroids structures (this work) (see also **Fig. S9** and **Table S3**). Heterogeneity is indeed always a valid concern for native samples, but exactly the same applies to the deglycosylated endogenous hTG purified from thyroid glands of patients with goitres and (to a lesser extent) to the recombinant hTG used in Coscia *et al.*

We used the ‘traditional’ domain naming as represented in the Uniprot database for entry UniProtKB - P01266 (THYG_HUMAN), including Thyroglobulin type-1-11, Type II, Type IIIA/B and Cholinesterase-like (ChEL), that were not disrupting secondary structure elements. No new names were used according to the recommendation of reviewer 1 and the new **Table S3** facilitates easier comparison of structures, given domain names and structure zones. As mentioned above, the only terms we use to describe our hTG structure in **Fig. 1** are the ‘foot’ and the ‘wing’ as a structural region of interest for native, glycosylated hTG from healthy thyroids (and which can also be identified in the structures of Coscia *et al.*).

This manuscript utilizes mass spectrometry to analyze posttranslational modifications in hTG. However, this methodology is not described in the corresponding section (nor elsewhere in the manuscript). From Table 1, I assume the authors used LC-MS. Please include details on how mass spectrometry was performed and the raw data in each of the corresponding sections (the description of the methodology, the proteases used and where they map in the Tg monomer, the equipment used, the chromatograms).

We thank the reviewer for pointing this out. We now include LC-MS methods in the ‘Methods’ section. LC-MS data used to generate tables and figures will be provided as a .xlsx source data file (SourceData_PTMs_Ionscore40_final.xlsx).

The raw data can be accessed on the proteomics server under the accession codes PXD029013 and 10.6019/PXD029013 and has been added to that section in the main text.

Line 163. Replace "coupling is proposed" for "coupling occurs".

We agree with the proposed wording and we changed the text accordingly.

Line 167. "...donor/acceptor site for TSH production". The concept is wrong. Please replace for "...donor/acceptor site for thyroid hormone production".

Indeed, there is an abbreviation error in line 167. We changed the text to mention the correct thyroid hormone abbreviation, TH, introduced before. It now reads "(...) donor/acceptor site for TH production(...)".

Line 176. "...acceptor at Y2766, specific for T4 and T3 synthesis) respectively." Please replace for "...acceptor at Y2766, the main site of T3 formation)."

We agree with the proposed wording and we changed the text accordingly.

Line 172-173. "The iodination pattern may be inconsistent between different particles." I strongly recommend the authors to perform further experiments to determine iodination sites in TG with an homogeneous source of iodinated hTG to clarify this. The iodination of Tyr and utilization of hormonogenic sites varies according to the different factors, e.g. the availability of iodine, the stimulus of TSH.

The iodination homogeneity of our sample reflects that of the native protein extracted from human thyroid tissue. To obtain a highly homogeneous sample in regard to iodination biochemical modifications, such as *in vitro* iodination, would be required. In order to preserve the native iodination pattern we decided not to perform any biochemical modification affecting tyrosines on the sample. We cannot reliably identify any iodination in our cryo-EM structure even though we detected many of these iodination sites by MS, therefore we may conclude: 1) the iodination pattern is very likely to be heterogeneous in our sample due to the above mentioned reasons causing iodinated tyrosines to be poorly defined in the density map, 2) the iodination pattern itself does not cause major structural changes in hTg since our structure is very similar to those of Coscia *et al.* (both recombinant glycosylated and endogenous, deglycosylated (from goiters)).

Performing additional experiments to characterize a homogeneous sample is out of the scope of this work and it is, in our view, contrary to the idea of characterizing a native hTg sample.

Line 178. Site E is not represented in Fig 6. Please rectify/move the following "(Fig. 6, Table1)". Was site E determined by cryo-EM? Please clarify and if so, please include in Fig. 6.

For sake of simplicity and due to the highly hypothetical nature of site E we reworked the text which now does not mention this site (**Fig. 6, Table 1**). **Fig. 6** remains therefore as previously presented.

Please clarify the distances between Tyr24 and all proposed donor sites to Tyr24 that are mentioned in this manuscript: Tyr24-Tyr149; Tyr24-Tyr234; Tyr24-Tyr258

We thank the reviewer for pointing out that this important information was missing in the text. Since the first modelled residue in our work is Pro30, we use it as a proxy to estimate the distance between the potential donor sites and Tyr24, which are as follows:

Distance between Pro30.A CA and Tyr149.A CA: 20.6 Å
Distance between Pro30.A CA and Tyr234.A CA: 19.9 Å
Distance between Pro30.A CA and Tyr258.A CA: 35.3 Å

We have also updated the text in the corresponding section (“Mapping of hTg’s hormonogenic sites”) to include this information.

Line 202. “We did not observe iodination of Y1310 nor Y108 and our data does not support those residues as a D-site” This is not consistent with what is shown in Fig. 6. Please clarify and amend accordingly in the text / Fig. 6.

We acknowledge the inconsistency pointed out by the reviewer. The text has been modified and it now reads “We did not observe iodination of Y1310 nor Y108 in the MS data nor in the cryo-EM density map.”, therefore not suggesting that this site does not exist but simply stating we did not gather evidence for iodination at the mentioned tyrosines in our study. **Fig. 6** and its legend are now modified so the term “canonical site” is employed conveying the idea that such sites were identified and studied in prior work.

Line 202-203. Need careful revision of the paper cited (25. Berg G et al. 2020), which I was not able to find online. I recommend incorporating the following bibliography and revise the hypothesis related to in vivo/ in vitro utilization of site D: *Biochem Biophys Res Commun.* 2014 Jul 18;450(1):488-93. doi: 10.1016/j.bbrc.2014.05.144. Epub 2014 Jun 6; *J Biol Chem.* 2011 Jan 7;286(1):259-69. doi: 10.1074/jbc.M110.159483. Epub 2010 Oct 26.

Indeed, there was a citation error in the corresponding line, this should cite Coscia *et al.* 2020. We changed the text to include the correct citation as well as both suggested citations at the end of this paragraph.

Table 1. It is not clear what “iodination” is referring to, please indicate MIT or DIT accordingly.

Table 1 was modified and Monoiodotyrosination (MIT) and Diiodotyrosination (DIT) were mentioned to make the table more clear.

Please include a legend to Table 1. Indicate that the numbering of the amino acid position in TG refers to TG including the signal peptide and cite the reference sequence.

A legend was included in **Table 1** highlighting that the numbering system of amino acid residues refers to human thyroglobulin including the signal sequence, indicating the Uniprot sequence (P01266) used.

Line 227-258. Clarify the methodology utilized to determine hTG glycosylation and other modifications as well as the source of hTG. Please add detailed description as mentioned in my previous comment #8.

As also stated above, LC-MS methods are now added in the 'Methods' section. LC-MS and data used to generate tables and figures are provided as a .xlsx source data file (SourceData_PTMs_Ionscore40_final.xlsx).

How was the mapping of the TG proteolysis sites created? If this is a predictive informatic analysis this should be indicated in the text and in Figure legend #3.

The mapping of non-tryptic cleavage sites was carried out experimentally using MS. The information that the cleavage sites were experimentally determined was added in the figure legend of **Fig. 3** and source data can be found in the provided MS source data .xlsx file (SourceData_PTMs_Ionscore40_final.xlsx).

Line 253. "We considered all the nonsense and missense mutations reported previously (1. Citterio et al., 2019)" Is this sentence referring to the (nonsense and missense) mutations in TG responsible for congenital hypothyroidism? If so, the data in this manuscript represented in Figure 7 should be updated to the most recent compilation of the hTG mutations responsible for congenital hypothyroidism that was published this year -- please refer to Mol Cell Endocrinol. 2021 May 15;528:111227. doi: 10.1016/j.mce.2021.111227. Epub 2021 Mar 6. Additionally, please modify the sentence "We considered all the nonsense and missense mutations responsible for congenital hypothyroidism reported previously" and add the proper citation.

Thank you for the relevant comment. We acknowledge the work in the given citation as an updated and more extensive compilation of missense and nonsense hTg mutations. We repeated the analysis/plotting of the referred mutations in the hTg structure and arrived at the same conclusion: we describe 5 regions with high mutation density (referred as mutation clusters), furthermore, the location of these sites is the same as presented in the previous submitted text. The reference of Citterio *et al.* 2021 has now been included. The number of nonsense and missense mutations reported and visualized in the structure has been updated in the text under Section 'Mapping of nonsense and missense mutations'. The number of mutations has increased to 42 nonsense mutations (in the histogram) out of which 37 are visualized on the structure in **Fig. 7**, as 130 missense mutations (in the histogram) out of which 120 are pictured on the structure. The reason that not all mutations can be displayed on the structure is that some (flexible) regions in the structure have not been modeled. We have modified the title of the figure legend of **Fig. 7** accordingly.

Line 81. "hTG" replace to "human TG (hTG)"

The abbreviation "hTg" present at line 81 was introduced earlier, at line 75 of the original manuscript: "Type 1 repeats occur 11 times in the human Tg (hTg)(...)". We respectfully disagree with this minor change and would propose not to alter the text.

Figure legend 1. Line 453. Indicate that you refer to the primary hTG structure.

Thank you, we have changed the figure legend accordingly.

Figure legend 2. Clarify what the white/grey ribbon represents.

Non-type 1 repeats are colored in light grey in **Fig. 2**. This information has now been added to the figure legend.

Figure legend 4. Indicate the meaning of each color in the ribbons. What does yellow color mean?

We added the information of our type 2 repeat representation as pink, light and dark purple ribbon and their connectivity through DSBs and their sulfur atoms (colored yellow) in the figure legend of **Fig. 4**.

Figure legend 5. Provide further explanation of the representation. What is the upper panel representing? How is it different from the lower panel?

The legend of **Fig. 5** has been reformulated to clarify the representation on the top and bottom panel.

Figure legend 6. Specify the amino acid number of the acceptor sites. Clarify the methodology by which acceptor sites B and D were identified.

We have now specified the relevant residue numbers in the legend of **Fig. 6**. We have also reformulated the figure legend to clarify how these acceptor sites were identified.

Supplementary figure S4: Indicate the meaning of FSC (Fourier shell correlation).

The figure legend has been updated accordingly.

Video. It would be beneficial to match the (letter) color of the names of each of the Tg zones or sites and the respective Tg zone or site in the Tg 3D structure.

We modified the video so that the sliding titles have accompanying boxes color matched to the highlighted domains/sites in the hTg structure.

Reviewer #2

The paper by Taylor and co-workers describes the structure of the native human thyroglobulin using cryo-EM. The paper should

be published provided that the points I indicate below are resolved:

We thank the reviewer for their detailed review and appreciative words.

Provide a comparison to the structure recently published by Coscia et. al., an RMSD and a figure highlighting the major differences found and what the new structures add to the overall picture.

We have included a new supplementary figure (**Fig. S9**) comparing the hTg model obtained in our study with that of Coscia *et al.* based on RMSD. Although the two models are overall similar to each other, there are indeed areas where they deviate considerably. For these areas we also show our cryo-EM density and its respective threshold level for interpretation, as our modelling is often backed up by densities for large side chains such as Arg, Trp or Tyr and we stand by it. Other sources of deviation are for example loops and gaps in areas where the cryo-EM map was poorly resolved in both studies, and therefore remain to be settled in future studies with higher-resolution data. Furthermore, our native hTG sample was not deglycosylated, whereas that of Coscia *et al.* was.

There is no information on model quality and there should be a table with the refinement and validation statistics for the model either in the main text or supplementary info.

Thank you for pointing this out. We have added a new **Table S1** summarizing all information on data collection, image processing and model statistics.

The clash score is quite high and the large number of clashes should be addressed.

We thank the reviewer for pointing this out. Indeed, the number of clashes in our model was rather high. After careful inspection of these clashes and further real-space refinement in PHENIX, we managed to bring the clashscore down to 5. In particular, we found useful to increase the weight of nonbonded interactions (repulsion terms) following this advice on the phenixbb mailing list: <https://phenix-online.org/pipermail/phenixbb/2018-February/023699.html>.

The model was manually inspected after refinement to ensure agreement with our previous interpretation and a reasonable fit to the experimental density. The all-atom RMSD between the old and the new model is only 0.3 Å.

In all figures, the authors should state and show the densities at 1 σ contour level. This is specially important in Figure S2

We respectfully disagree with the reviewer in this comment. There is no standard way to normalize maps in cryo-EM, such that the threshold level relevant for interpretation of the map can widely vary across samples, data collection sessions and reconstruction software, among other factors. This is unlike X-ray crystallography, where intensities are scaled in relation to the number of incident photons on a unit cell and therefore resulting electron density maps are normalized in terms of σ (RMSD) levels. We provide below a figure of our

hTg map at 1σ level to demonstrate this would not be informative for inclusion in the manuscript.

We do agree with the reviewer, however, that in **Fig. S5** it makes sense to render the map at the same σ level to illustrate the differences between the consensus and the composite map at the “foot” and “wing” regions (shown at 7σ in this case). We have therefore updated this figure and the corresponding legend. Furthermore, we also now provide σ levels in **Fig. 6** and **Fig. S9** as they help assess the quality of the densities, an important concept for these figures.

The corresponding half-maps, masks and the raw maps should be deposited to EMDB together with the PDB entry.

Absolutely. The half-maps, mask, final map and model have now been deposited in addition to the PDB model (accession code 7B75) and the EMDB main map (accession code 12073).

Two PDB reports were submitted but correspond to the same entry, please submit one only.

Thank you for pointing this out, the PDB report has now been submitted only once.

Given the importance highlighted in the manuscript of the glycosylation sites, it would be informative to include a figure showing the density of a representative site.

The manuscript now includes **Fig. S8** depicting 7 of the modelled glycosylations and MS-confirmed N-linked glycosylation sites.

In figure S1 the local resolution estimate for wing and foot should be provided.

We updated the flowchart provided in **Fig. S2** to include the resolution estimates for the wing (3.8 Å) and the foot (4.1 Å) after local refinement in cryoSPARC.

The dip in the FSC curve should be addressed.

We assume the reviewer refers to the dip, or slight valley, centered at ~ 6 Å, which is more pronounced in the unmasked FSC curve (see FSC curve comparison below):

A possible explanation would be the high variation of local resolution across the map. The regions highlighted with dashed lines in the figure below clearly include areas of resolution limited to around 6 Å, comparing local resolution of the consensus and composite map (see below). When the global FSC is calculated, an average correlation over the whole map in Fourier space is obtained. That is, parts that are less well resolved are mixed together with parts that are resolved to high resolution. It is therefore to be expected that given our map has several regions limited to ~ 6 Å resolution, that the global FSC would be dragged down in this resolution range. The FSC curve then rises again at ~ 5 Å due to the contribution of the high resolution core of the map, in particular, due to beta-sheets that start to scatter strongly at this range.

In summary, we attribute this “dip” in the global FSC curve to the local resolution variation of the map. While more common in structures of membrane proteins due to the disordered lipid belt, this phenomenon also occurs in cryo-EM of soluble proteins, as can be observed for example in the following references which describe particularly flexible structures:

<https://doi.org/10.1016/j.cell.2021.05.011> (Figure S1)

<https://doi.org/10.7554/eLife.70871> (Figure 1 Supplement 1)

<https://doi.org/10.1038/s41594-021-00637-y> (Extended Data Fig. 2)

<https://doi.org/10.1038/s41594-021-00632-3> (Extended Data Fig. 1)

<https://doi.org/10.1038/s41586-021-03906-4> (Extended Data Fig. 7)

Figure S4 the FSC for the consensus map should also be presented. Ln 93 The resolution from the structure published by Coscia et al. should be included. L101 Where are the residues that could not be shown? Ln 123 The estimated local resolution for the foot should be stated. Ln 324 Include the initial model used or if constructed ab initio describe the method. Ln 346 Title is in all caps

We have included the FSC curve for the consensus map in **Fig. S4**. Furthermore, we have included the resolution of the structures from Coscia *et al.* in the corresponding paragraph. Some residues could not be modelled because of high local resolution variation of the map due to structural flexibility. This is now acknowledged at the beginning of the results section.

The estimated local resolution of the wing (3.8 Å) and the foot were included (4.1 Å). More details of the model building are now provided in the methods. The title of the Pharoah *et al.* citation was corrected.

Reviewer #3

Adaixo et al. present the cryo-EM structure of native human thyroglobulin (Tg). Tg is an important molecular complex that plays a key role in the synthesis of thyroid hormones. In addition to structure determination, the authors have performed mass spectrometric analysis of their specimen. From these data, the authors gain insight into the residues that are critical for hormone synthesis, and they identify protein residues that are post-translationally modified by iodination, glycosylation, sulfonation, and phosphorylation. Overall, these results are in good agreement with a study that previously reported the structure of native and recombinant human thyroglobulin (Coscia et al., reference 23), though minor differences exist.

One of the major concerns with the study presented by Adaixo et al. is that it mostly ignores the Coscia et al. paper, likely aiming not to compromise the novelty of the findings, even though the Coscia et al. study was published more than one year ago. This not only raises issues with appropriate acknowledgment of prior work, but also leaves the comparison of the new and previous results to the reader, diminishing the utility of the study under consideration.

In its present form, I cannot recommend the manuscript under consideration for publication. The manuscript should be revised to account for the current state of knowledge in the field.

We thank the reviewer for the detailed assessment and valuable comments.

The structure and experiments were carried out independently and at about the same time as Coscia *et al.* 2020, but due to the COVID-19 pandemic the publishing process of our data has been significantly delayed. However, unlike Coscia *et al.*, our structure and data represents native, fully glycosylated human thyroglobulin extracted from healthy human thyroid glands and adds valuable insight about iodination state, glycosylation sites, proteolytic cleavage sites. Furthermore, our structure is of somewhat better global resolution and in particular, various flexible regions are better resolved in our structure. The structure of native, fully glycosylated structure of healthy thyroids and the associated MS data are completely complementary to those of those of recombinant hTg and deglycosylated hTg from patients with goiters that was published by Coscia *et al.* 2020. However, to aid the reader in having the best overview of the current state of the art, we worked on more extendedly describing the data presented in Coscia *et al.*

52: "structure conserved among vertebrates" - maybe better use "architecture" or "domain organisation" instead of "structure" unless other structures are known.

We agree with the proposed wording and changed the text to: "Its architecture is conserved among vertebrates."

90-94: The introduction needs to be rewritten to properly acknowledge and reference prior work by Coscia et al., a paper published in February 2020, i.e. more than one year ago. Doing so will better inform the reader about the current state of the field and the new insight provided by the work under review. The wording suggesting that Coscia et al. reported only the structure of recombinant Tg is inaccurate and should be changed because Coscia et al. reported the structure of human Tg extracted from thyroid glands as well.

We thank the reviewer for their valuable suggestions that have as well been proposed by the other reviewers (see above). We have modified the paragraph accordingly.

204-208: Residues Y883, Y2637, and Y2640 seem to be buried in the protein core and unlikely to be accessible as donors/acceptors. Overall, this section remains unconvincing, particularly because the lack of in vivo iodination was used in the preceding paragraph to discount site D, and when also considering the biochemical data in Coscia et al. that indicate the sites A-D account for all hormone production (at least in their system).

After careful consideration we agree with the reviewer that this section is too hypothetical. We have therefore chosen to omit it from the manuscript.

238/239: "... that have not been described in previous biochemical studies" - some of these were identified in the structural study by Coscia et al.; while factually correct, this claim appears not to fully convey the current state of knowledge in the field.

We reformulated the text in these lines to include the structurally observed glycosylations by Coscia et al. 2020.

Figure 1: It is unclear whether introduction of another new set of domain names is beneficial to the reader. The authors may want to reconsider.

We understand the reviewer's point and the reasoning behind it. As also mentioned previously, we have now included **Table S1** in the manuscript which addresses the differences in domain boundaries/names between our study and others. We opted for using the conventional names for all domains. The most striking difference between our naming/boundaries scheme and others is the assignment of the so called "linker" region as an insertion within hTg type-1 5 repeat and not as a domain *per se*.

Figure 2: Ribbon representation could use thicker ribbon to make repeats 1.6-1.9 more easily distinguishable against the background and from each other.

We thank the reviewer for this suggestion. A new version of this figure with thicker ribbon representation was provided.

Figure 6: The figure shows molecular models of sites B and D, including side chains. Densities supporting these assignments should be shown either directly in this figure or in a supplementary figure. The assignment appears reliable for site B. However, the panel for site D shows residues that are either not built at all in the molecular model (Y108) or whose location is not very well defined due to poor density in this region (Y1310). It is important to convey the level of accuracy and reliability with which these important sites could be modelled to the readers of the paper, preferably in the figure legend and/or in the main text.

We have updated **Fig. 6** to include the experimental density. Residues not modelled are now not shown, but their approximate position indicated. Besides the density itself, we also show the threshold level (in sigma-levels) relevant for visualizing each site, which should aid the reliability evaluation of our assignments.

Methods: The paper lacks appropriate validation of the cryo-EM-derived molecular model (e.g. refinement statistics, model-map FSC). These need to be added.

Thank you for pointing out this omission. We have now added a new **Table S1** summarizing all information on data collection, image processing and model statistics.

REVIEWERS' COMMENTS

Reviewer #1 (Remarks to the Author):

The authors have been responsive to previous criticism and the manuscript has improved. However, there are still concerns to be addressed.

Major comments:

1) Table S3 requires modifications: As the authors mentioned, the C-terminal end of human thyroglobulin (hTg) is not visible in its structure. It is important to note that it has been published that the hTg region that bears homology to the acetylcholinesterase (ACHE) corresponds to hTg aminoacidic positions 2187–2728; and there is a C-terminal unique tail afterwards (in the C-terminal end of Tg). This Tg unique tail is not homologous to the ACHE. Please modify the last 2 rows of table S3 accordingly.

2) From Table 1, it appears that phosphorylation at hTg S2740 is a novel finding. However, phosphorylation of hTg at site 2740 has been already described <https://www.phosphosite.org/proteinAction.action?id=19112&showAllSites=true> . Thus, table 1 as well as the corresponding discussion of this result require modifications.

Minor comments:

1) In the light of the recently published manuscript of the structure of natively iodinated bovine thyroglobulin (Acta Crystallogr D Struct Biol. 2021 Nov 1;77(Pt 11):1451-1459. doi: 10.1107/S2059798321010056. Epub 2021 Oct 29) and given that Tg has a highly conserved structure throughout vertebrates, I strongly suggest comparing the iodinated/hormonogenic sites found in this paper to the ones found in the current manuscript. This will add value to the current manuscript (by highlighting what is novel) and clarity of the concepts for the readers.

2) Please clarify which are the hTg Cys that were not found to be involved in disulfide bond formation in this work by adding this information in a newly modified Table S2 or a new supplemental table. Additionally, please discuss why the authors believe these were not seen in the Tg structure described in this work. Do this information about the disulfide bond formation in Tg match what has just been published regarding the structure of natively iodinated bovine thyroglobulin (Acta Crystallogr D Struct Biol. 2021 Nov 1;77(Pt 11):1451-1459. doi: 10.1107/S2059798321010056. Epub 2021 Oct 29)?

3) "Other" seems to be duplicated in the abstract: "as well as many other other post-translational modifications and novel proteolytic cleavage sites"

4) Main body: "Dimeric Tg is secreted to the follicular cavity and iodinated to different extents at specific tyrosine residues, a process modulated by the dietary iodine intake". Please reformulate the sentence to make it clear to the reader that the main factor regulating thyroid hormone synthesis (at the thyroid level) is the stimulation of TSH receptors.

5) The following sentences from the discussion section need attention and the corresponding amendments. "We modelled 2 of the expected 4 hormonogenic sites, namely site B and site C. Acceptor tyrosines of sites A and C are located at the extreme N and C terminus of the hTg chain and could not be modelled likely due to inherent flexibility of these regions."

Reviewer #2 (Remarks to the Author):

The authors have answered essentially all my concerns. I find the paper now suitable for publication, and compliments for the nice work!

Reviewer #3 (Remarks to the Author):

Adaixo et al. have provided a revised - and considerably improved - manuscript in which they have addressed the comments of three reviewers. My comments have been satisfactorily addressed and I have no remaining major concerns.

Minor comments:

Lines 114-116: I suggest removing "has long" and moving "recently" to after "revealed".

Line 238/239: Figure S3 shows that particles show multiple views but also that they are clearly not "randomly oriented".

Point-by-point response to reviewers' comments

“Cryo-EM structure of native human thyroglobulin” (Adaixo *et al.*, NCOMMS-21-08694)

We would like to thank all the reviewers for their careful re-reading of the manuscript and are very pleased all of them agree our study has much improved. In this revision we now address the points raised by the reviewers.

Reviewer #1

The authors have been responsive to previous criticism and the manuscript has improved. However, there are still concerns to be addressed.

Major comments:

1) Table S3 requires modifications: As the authors mentioned, the C-terminal end of human thyroglobulin (hTg) is not visible in its structure. It is important to note that it has been published that the hTg region that bears homology to the acetylcholinesterase (ACHE) corresponds to hTg aminoacidic positions 2187–2728; and there is a C-terminal unique tail afterwards (in the C-terminal end of Tg). This Tg unique tail is not homologous to the ACHE. Please modify the last 2 rows of table S3 accordingly.

We thank the reviewer for bringing this up and have modified the last two rows of Supplementary Table 2 (former Table S3) accordingly.

2) From Table 1, it appears that phosphorylation at hTg S2740 is a novel finding. However, phosphorylation of hTg at site 2740 has been already described
<https://www.phosphosite.org/proteinAction.action?id=19112&showAllSites=true> . Thus, table 1 as well as the corresponding discussion of this result require modifications.

Thank you for highlighting the known modification of S2740. We have changed Table 1 accordingly.

Minor comments:

1) In the light of the recently published manuscript of the structure of natively iodinated bovine thyroglobulin (Acta Crystallogr D Struct Biol. 2021 Nov 1;77(Pt 11):1451–1459. doi: 10.1107/S2059798321010056. Epub 2021 Oct 29) and given that Tg has a highly conserved structure throughout vertebrates, I strongly suggest comparing the iodinated/hormonogenic sites found in this paper to the ones found in the current manuscript. This will add value to the current manuscript (by highlighting what is novel) and clarity of the concepts for the readers.

We prepared a comparative table, Supplementary Table 4: Comparison of Modification Types and Sites in hTG and bovine TG. Kim *et al.* structurally observe three tyrosines with additional densities: Y24 (T₄), Y2041 (DIT), Y2575 (T₄). For Y24 we observe the same thyroxine formation by mass spectrometry, Y2041 is not conserved and for hTG Y2573 (corresponding to bovine TG Y2575) we see modifications to MIT and DIT by mass spectrometry. Furthermore, Kim *et al.* observed that four tyrosine residues, Y108, Y149, Y1395 and Y2542, which had clearly defined main chain density, lacked discernable side chain densities, indicating their modification to dehydroalanine. In our structure, Y108 could not be modeled, Y149 shows side chain density, bovine TG Y1395 is not conserved and hTG Y2540 (Y2542 in bovine TG) shows clear side chain density. We have added a paragraph with reference to Kim *et al.* in the text to introduce these results (as well as those concerning the cysteine residues, see further) to the reader.

2) Please clarify which are the hTg Cys that were not found to be involved in disulfide bond formation in this work by adding this information in a newly modified Table S2 or a new supplemental table. Additionally, please discuss why the authors believe these were not seen in the Tg structure described in this work. Do this information about the disulfide bond formation in Tg match what has just been published regarding the structure of natively iodinated bovine thyroglobulin (Acta Crystallogr D Struct Biol. 2021 Nov 1;77(Pt 11):1451-1459. doi: 10.1107/S2059798321010056. Epub 2021 Oct 29)?

We thank the reviewer for the valuable comment and prepared the requested table, Supplementary Table 3 (which also contains the information previously stated in Table S2). This supplementary table contains 61 possible disulfide bridges, including disulfide bridge C1249 – C1281 that was not yet annotated on Uniprot for human thyroglobulin but was observed in the bovine structure. These 61 potential disulfide bridges make up 122 of the 123 cysteines, the remaining C15 is located at the N-terminus and has not been modeled in our structure or any other structure. In our structure, we observe 57 disulfide bridges. The four non-observed disulfide bonds contain three for which one of the putative bonded cysteines is not modeled (C1215 – C1264, C1306 – C1356 and C1639 – C1662 (the latter pair (bovine TG C1642 – C1665) was modeled as close but non-disulfide bond-forming in Kim *et al.*)). The fourth non-observed disulfide bond (C1733 – C1835) was modeled as a disulfide bond by Coscia *et al.* but not by Kim *et al.* (bovine TG C1736 – C1837). The density in this region is unfortunately not very good for either the map of Coscia *et al.* or our own map, preventing a clear conclusion.

Furthermore, Kim *et al.* could observe additional pairs of cysteine residues that were in close proximity, but which did not form disulfide bridges, which they state they can do because of the high resolution of their map. In our map, the densities are consistent with the modelled formed bridges, but the overall resolution of our hTG structure is a bit lower than that of the bovine TG structure, therefore we prefer to stand by the more conservative assumption that they are bonded in hTG. We made a short note of this in the text at the end of the paragraph discussing Kim *et al.*, referencing Supplementary Table 3.

3) "Other" seems to be duplicated in the abstract: "as well as many other other post-translational modifications and novel proteolytic cleavage sites"

The sentence has been corrected in the abstract.

4) Main body: "Dimeric Tg is secreted to the follicular cavity and iodinated to different extents at specific tyrosine residues, a process modulated by the dietary iodine intake". Please reformulate the sentence to make it clear to the reader that the main factor regulating thyroid hormone synthesis (at the thyroid level) is the stimulation of TSH receptors.

In order to improve readability, this information is now provided shortly after the suggested sentence:

TG is simultaneously a precursor for thyroid hormone (TH) biogenesis, the synthesis of which is itself modulated by thyroid-stimulating hormone (TSH) signaling (Citteri et al., 2019), and the carrier protein responsible for iodine storage in the follicle colloid.

5) The following sentences from the discussion section need attention and the corresponding amendments. "We modelled 2 of the expected 4 hormonogenic sites, namely site B and site C. Acceptor tyrosines of sites A and C are located at the extreme N and C terminus of the hTg chain and could not be modelled likely due to inherent flexibility of these regions."

We have corrected the sentence to clarify that we have actually modelled sites B and D (not C):

We modelled 2 of the expected 4 hormonogenic sites, namely site B and site D. Acceptor tyrosines of sites A and C are located at the extreme N- and C- termini of the hTG chain and could not be modelled likely due to inherent flexibility of these regions.

Reviewer #2

The authors have answered essentially all my concerns. I find the paper now suitable for publication, and compliments for the nice work!

Thank you!

Reviewer #3

Adaixo et al. have provided a revised - and considerably improved - manuscript in which they have addressed the comments of three reviewers. My comments have been satisfactorily addressed and I have no remaining major concerns.

Thank you!

Minor comments:

Lines 114-116: I suggest removing "has long" and moving "recently" to after "revealed".

The sentence has been amended as suggested.

Line 238/239: Figure S3 shows that particles show multiple views but also that they are clearly not "randomly oriented".

We modified the text to the following:

The cryo-EM images of plunge-vitrified hTG solution displayed monodisperse particles in different orientations with size and shape consistent with previous observations in negative stain preparations¹⁷.